# Autocrine insulin pathway signaling regulates actin dynamics in cell wound repair

Mitsutoshi Nakamura[1⦿], Jeffrey M. Verboon[1⦿], Tessa E. Allen[1], Maria Teresa Abreu-Blanco[1], Raymond Liu[1], Andrew N. M. Dominguez[1], Jeffrey J. Delrow[2], Susan M. Parkhurst[1]*

1 Basic Sciences Division, Fred Hutchinson Cancer Research Center, Seattle, WA, United States of America, 2 Genomics Shared Resource, Fred Hutchinson Cancer Research Center, Seattle, WA, United States of America

⦿ These authors contributed equally to this work.
* susanp@fredhutch.org

**Data Availability Statement:** The microarray datasets are available at GEO (NCBI Gene Expression Omnibus) under accession numbers: GSE39481, GSE39482, and GSE39483.

## Abstract

Cells are exposed to frequent mechanical and/or chemical stressors that can compromise the integrity of the plasma membrane and underlying cortical cytoskeleton. The molecular mechanisms driving the immediate repair response launched to restore the cell cortex and circumvent cell death are largely unknown. Using microarrays and drug-inhibition studies to assess gene expression, we find that initiation of cell wound repair in the *Drosophila* model is dependent on translation, whereas transcription is required for subsequent steps. We identified 253 genes whose expression is up-regulated (80) or down-regulated (173) in response to laser wounding. A subset of these genes were validated using RNAi knockdowns and exhibit aberrant actomyosin ring assembly and/or actin remodeling defects. Strikingly, we find that the canonical insulin signaling pathway controls actin dynamics through the actin regulators Girdin and Chickadee (profilin), and its disruption leads to abnormal wound repair. Our results provide new insight for understanding how cell wound repair proceeds in healthy individuals and those with diseases involving wound healing deficiencies.

## Author summary

Organisms are constantly subject to damage by physiological and environmental stresses at the cell, tissue, and organ levels. While organisms have robust repair systems to survive from such damage, the underlying molecular mechanisms for these different scales of repair are different. Using microarray analyses and pharmacological assays with the *Drosophila* model, we examined the requirements for transcription and translation during cell wound repair. We find that translation, rather than transcription, is needed for the initial steps of cell wound repair. Transcription is required for the later steps of the repair process. We have successfully identified and verified 80 up-regulated and 173 down-regulated genes, most of which are new players in cell wound repair. A number of these genes function to regulate cytoskeleton dynamics at different steps of cell repair process.

**Funding:** This research was supported by National Institutes of Health grant GM111635 (to SMP). Cellular Imaging and Genomics Shared Resources used for this study are supported by National Cancer Institute Cancer Center Support Grant P30 CA015704.

**Competing interests:** The authors have declared that no competing interests exist.

Interestingly, a subset of these genes encode components of the insulin signaling pathway. While insulin signaling is required for tissue and organ wound repair, we find that a canonical insulin pathway is activated upon wounding in the repair of individual cells as well. Our results provide new insight for understanding how cell wound repair proceeds in healthy individuals and those with diseases involving wound healing deficiencies.

## Introduction

Numerous cell types in the body are subject to high levels of stress daily. These stresses—physiological and/or environmental—can cause ruptures in the plasma membrane and its underlying cytoskeleton, requiring a rapid repair program to avert further damage, prevent infection/death, and restore normal function [1–11]. Injuries to individual cells also occur as a result of accidents/trauma, clinical interventions, and disease conditions, including diabetes, skin blistering disorders, and muscular dystrophies, as well as in response to pore forming toxins secreted by pathogenic bacteria [12–17]. Repair of these cell cortex lesions can be particularly troublesome when occurring alongside these fragile cell disease states or in a non-renewing and/or irreplaceable cell type. Thus, the importance of cell cortex continuity and delineating the molecular mechanisms regulating cell wound repair is of considerable clinical relevance, and important for advancing our knowledge of the many critical cell behaviors and fundamental regulations underpinning normal biological events that are co-opted for this repair process.

Aspects of single cell wound repair dynamics have been studied in *Xenopus* oocytes, sea urchin eggs, Dictyostelium, mammalian tissue culture cells, and the genetically-amenable *Drosophila* syncytial embryo [2, 3, 18–24]. This repair is generally conserved among these organisms and occurs in four main phases (Fig 1A). In the first phase, the wound expands as the cell recognizes the membrane breach, releases resting membrane tension, and subsequently forms a membranous plug to neutralize any flux between the extracellular space and cytoplasm. Second, the cell constructs an actomyosin ring that underlies the plasma membrane at the wound edge. Third, the actomyosin ring translocates inward to draw the wound area closed. Mechanistic variations exist during this step wherein the actomyosin ring in some models translocates through actin treadmilling (actin simultaneously polymerizes at the inner edge and depolymerizes at the outer edge of the actin ring), while others use myosin II for sarcomere-like contraction (anti-parallel actin filaments are directed past each other in opposing directions) [3, 23–27]. In the final step of wound repair, the plasma membrane and the underlying cortical cytoskeleton are remodeled returning them to their pre-wounded composition and organization. The mechanisms deployed by the cell for this remodeling have not yet been delineated.

Previous studies have shown that $Ca^{2+}$ is required for the initiation of cell wound repair and serves as a messenger to trigger downstream processes such as transcription: release of internal and/or external $Ca^{2+}$ stores activates a number of intracellular pathways resulting in an uptick of gene expression [28–31]. Studies carried out in rat embryos and cultured bovine aortic endothelial cells showed a rapid increase in expression of the $Ca^{2+}$-responsive element containing c-Fos protein as a direct result of plasma membrane damage [28, 32, 33]. c-Fos, a component of Activator protein 1 (AP-1), serves as a transcription factor responsible for expressing a number of cytokines and growth factors required to drive the appropriate cellular responses necessary for epithelial (tissue) wound recovery [34–38].

Interestingly, though the *Drosophila* syncytial embryo functions under the developmental control of maternally-contributed mRNAs and proteins with minimal levels of zygotic

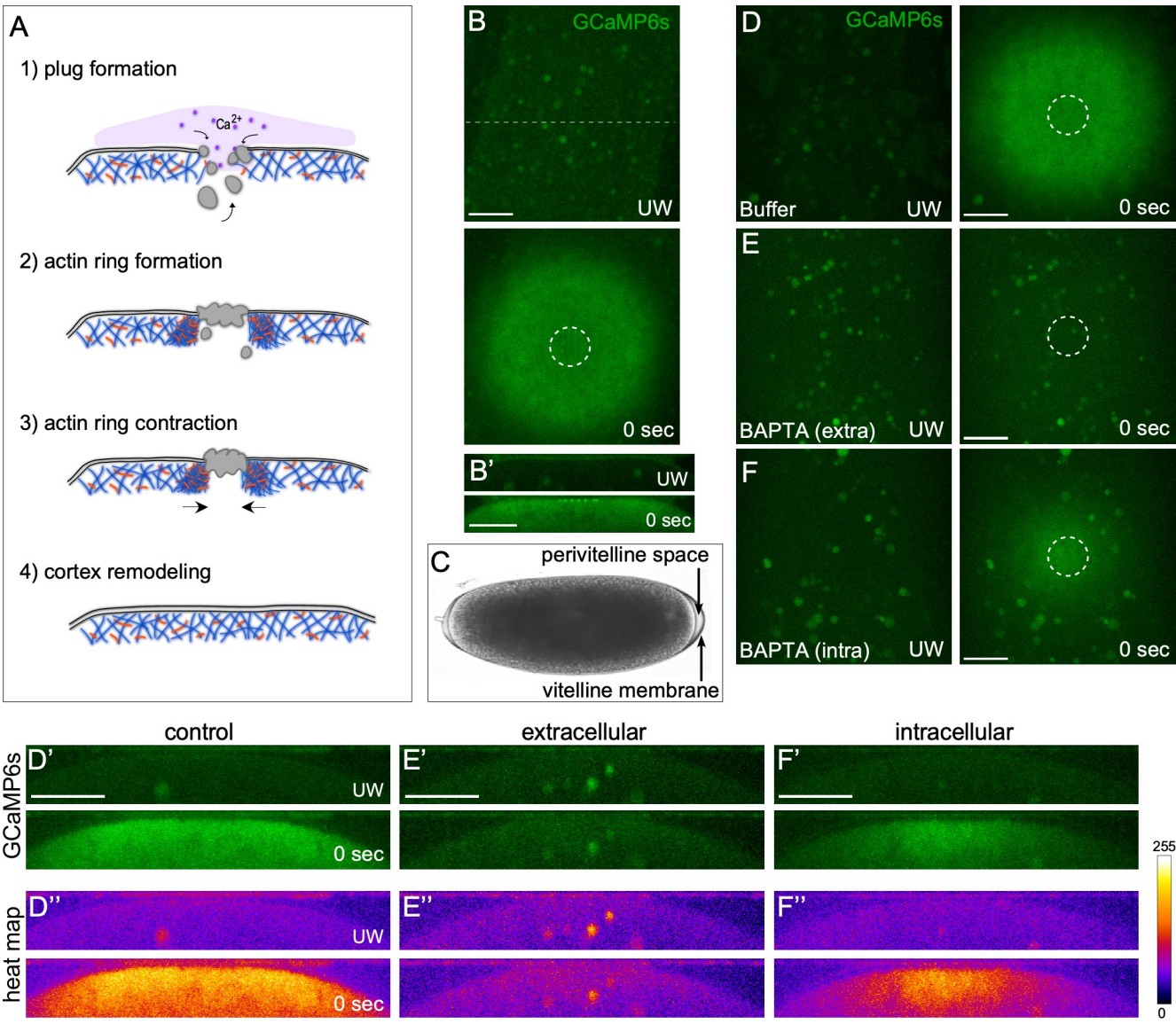

**Fig 1. Both intracellular and extracellular calcium sources contribute to the robust calcium response observed upon wounding in the *Drosophila* cell wound model.** (**A**) Schematic of the four major phases of cell wound repair. (**B-B'**) Confocal XY projections (B) and XZ projections (B') of calcium dynamics using time-lapse spinning disk microscopy in unwounded (UW) and immediately after laser wounding (0 sec) in *Drosophila* NC4-6 embryos expressing the GCaMP6s calcium reporter. (**C**) Bright field image of *Drosophila* syncytial (NC4-6 stage) embryo showing the vitelline membrane and extracellular perivitelline space. (**D-F**) Confocal XY projections of calcium dynamics in unwounded (UW) and immediately after laser wounding (0 sec) in *Drosophila* NC4-6 embryos expressing GCaMP6s following buffer injection (D), extracellular BAPTA injection (E), and intracellular BAPTA injection (F). (**D'-F'**) Confocal XZ projections derived from a 10 pixel width across the wound area in the embryo shown (C-F), respectively. (**D''-F''**) Heat map depicting the intensity of the calcium response in the confocal XZ projections shown in (C-F), respectively. Scale bars: 20 μm.

transcription, it is still able to immediately recognize and repair breaches to its cortex. Here we show that translation, rather than transcription, is required for the initial stages of repair in this cell wound repair model. Although transcription does not serve as a "start" signal, disrupting transcription leads to impaired repair in subsequent steps of the process. Using microarrays to assess gene expression changes post-wounding, we have identified 253 genes with a potential role in cell wound repair, indicated by changes in their expression—either up or down—in response to laser wounding. A subset of these genes were analyzed using RNAi

knockdowns to visualize spatio-temporal patterns that verified their involvement. Strikingly, we find that the canonical insulin signaling pathway is required for proper cell wound repair where it controls actin dynamics through the actin regulators Girdin (Hook-like protein family) and Chickadee (profilin). Thus, our study provides insight into the roles of transcription, translation, and insulin signaling in cell wound repair and provides new avenues for understanding how wound healing proceeds in healthy individuals and disease sufferers with wound healing impairments.

## Results

### Both intracellular and extracellular calcium stores are released upon wounding

The primary physiological cue for cell wound repair is thought to be entry of calcium into cells or release of internal calcium stores [1, 2, 5, 19, 39, 40]. This response is highly conserved among different cell wound repair models, but has not yet been examined with the *Drosophila* syncytial (nuclear cycle 4–6) embryo model. To determine if calcium influx occurs upon wounding, we wounded embryos expressing a constitutive calcium reporter (sqh-GCaMP6s) that is expressed under the control of the sqh promoter (see Methods) [41]. As expected, we observed a robust and immediate (<1 sec) response that spread 1–2 wound diameters from the cell cortex breach (Fig 1B-1B'; S1 Video).

The *Drosophila* model is somewhat unusual among cell wound repair models in that the syncytial embryo is enclosed in an impermeant vitelline membrane (Fig 1C). The space between the embryo plasma membrane and vitelline membrane (perivitelline space) contains extracellular matrix material and has been shown to contain $5 \pm 0.3$ mM calcium, compared to the internal embryo cytoplasm stores at ~90 nm calcium [42–44]. Importantly, our laser system allows us to wound the embryo without disrupting the enclosing vitelline membrane. To determine if the calcium response observed upon wounding is due to the external influx of calcium or the release of internal stores, we injected the specific calcium chelator, BAPTA, into either the perivitelline space (extracellular) or inside the embryo (intracellular) to deplete calcium. We then performed time-lapse imaging of calcium dynamics following laser wounding (see Methods) (Fig 1D-1F"; S1 Video). Interestingly, we find that both sources of calcium contribute to the calcium response upon wounding: inhibition of the extracellular calcium influx allowed only a small response from internal calium release over the same area as the wildtype calcium response (Fig 1E-1E"), whereas inhibition of the intracellular calcium release led to a weaker response centered on the region of the cell cortex breach (Fig 1F-1F").

### Assessment of transcriptional contribution to cell wound repair

To investigate the role of transcription in cell wound repair using the *Drosophila* syncytial (nuclear cycle 4–6) embryo model, we performed a microarray screen on full-length cDNA arrays to compare changes of gene expression between laser wounded and non-wounded states at two time points: immediately after wounding (0–5 minutes post-wounding (mpw)) and at the end of the repair process (~30 mpw) (Fig 2A). We found that at the immediate timepoint, wounded embryos exhibited no significant changes in their expression profiles when compared to their non-wounded counterparts (Fig 2B). Interestingly, the later timepoint, which was expected to identify any repair requirements post-initiation, showed significant changes of gene expression in both the up and down directions (Fig 2C). Using a false discovery rate of 0.05, we identified 253 genes with statistically significant changes: 80 that are up-regulated and 173 that are down-regulated (Table 1; S1 Table). The robustness of the

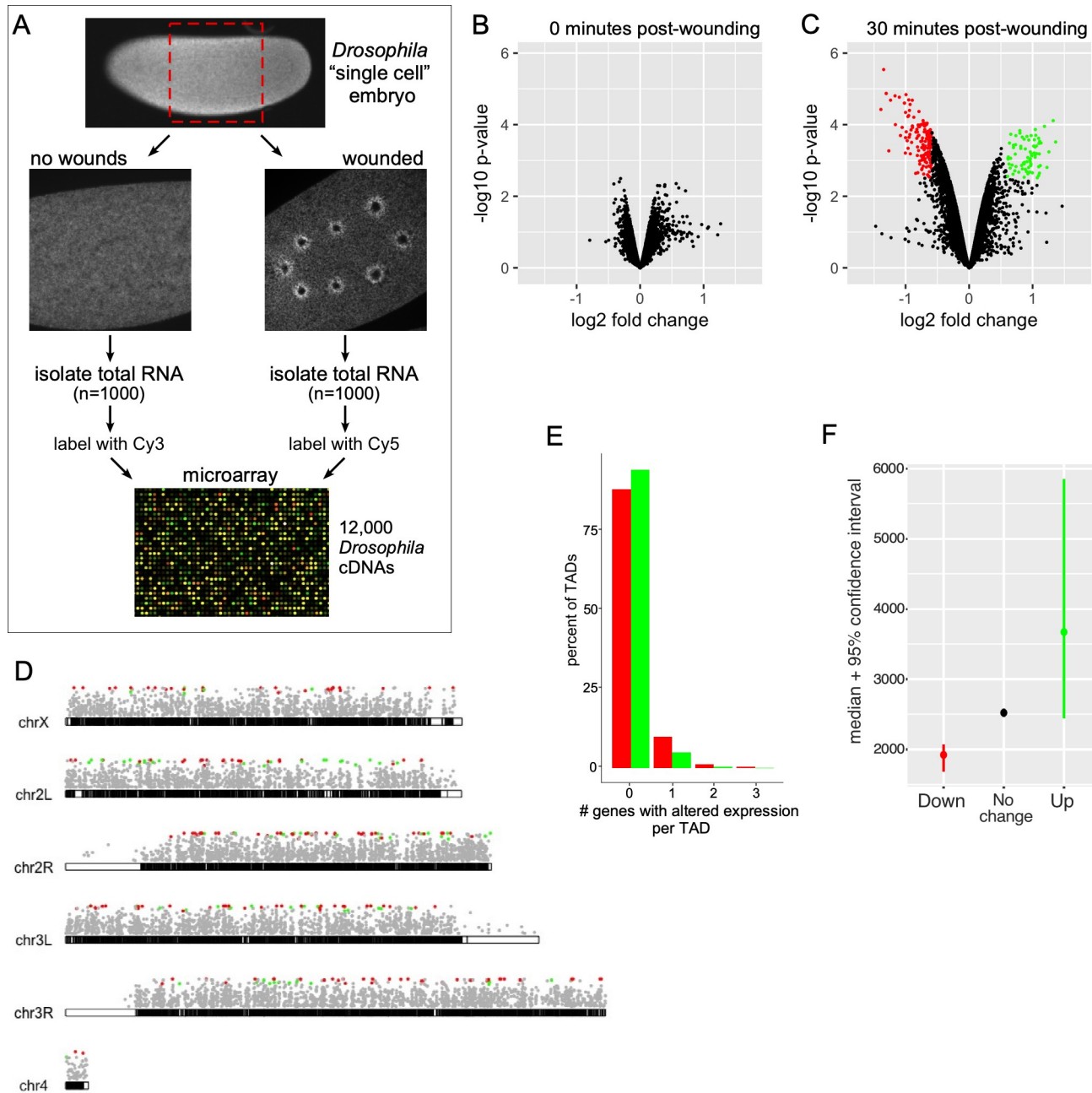

**Fig 2. Analysis of differential gene expression following wounding in the *Drosophila* cell wound repair model.** (**A**) Flow chart depicting the steps involved in microarray processing for examining the transcriptional response to cell wound repair. These analyses were performed for two-timepoints post laser wounding: immediate (0–5 mpw) and near completion (~30 mpw). (**B-C**) Volcano plots showing the differential gene expression for each of the two timepoints. Each dot represents a cDNA corresponding to its fold-change and p-value. Insignificant hits are depicted in black, whereas up-regulated and down-regulated genes are depicted in green and red, respectively. (**D**) *Drosophila* chromosome maps with each of the 4 chromosomes represented by euchromatic regions in black, heterochromatic regions in white, and with the left and right arms of chromosomes 2 and 3 depicted separately. Dots representing genes hits from the late-period microarray with significant up-regulated genes (green) and down-regulated genes (red) placed at their respective location within the genome. (**E**) Percentage of TADs containing the indicated number of up- or down- regulated genes per TAD. (**F**) Average gene size of significantly expressed genes from the ~30 min time point microarray. See S3 Table for numerical data.

**Table 1. List of top 16 Up- or 16 Down- regulated genes at t = 30 minutes.**

| FB Gene ID | Name | logFC | P-Val | Molecular/Biological function |
|---|---|---|---|---|
| FBgn0001257 | ImpL2 | 1.363 | 0.022 | - Insulin-like growth factor binding; Insulin signaling |
| FBgn0033855 | link | 1.324 | 0.022 | - Unknown; Involved in neurogenesis |
| FBgn0261560 | Thor | 1.257 | 0.026 | - EIF4E binding protein; Insulin signaling |
| FBgn0038071 | Dtg | 1.226 | 0.036 | - Unknown; Involved in gastrulation |
| FBgn0020300 | geko | 1.187 | 0.022 | - Unknown; Involved in olfaction |
| FBgn0263776 | CG43693 | 1.133 | 0.022 | - Amino acid transmembrane transporter |
| FBgn0038028 | CG10035 ** | 1.121 | 0.035 | - Unknown |
| FBgn0003731 | Egfr | 1.112 | 0.022 | - EGF receptor; Involved in growth regulation and development patterning |
| FBgn0000071 | Ama | 1.108 | 0.043 | - Immunoglobin-like protein domains; Involved in cell adhesion |
| FBgn0086910 | l(3)neo38 | 1.105 | 0.026 | - Zinc finger (C2H2-type); Regulation of transcription/chromatin silencing |
| FBgn0010109 | dpn | 1.100 | 0.023 | - basic Helix-Loop-Helix protein; Transcriptional regulation of sex determination and neurogenesis |
| FBgn0013272 | Gp150 | 1.096 | 0.036 | - Transmembrane glycoprotein; Regulates Notch signaling |
| FBgn0004143 | nullo | 1.095 | 0.022 | - Actin binding; Regulation of epithelial morphogenesis and actomyosin contractile ring assembly |
| FBgn0039283 | danr | 1.094 | 0.046 | - Homeobox domain; Transcriptional regulation of eye development and CNS formation |
| FBgn0028371 | jbug | 1.073 | 0.050 | - Filamin; Involved in cytoskeleton dynamics, PCP pathway, and mechanical stimulus response |
| FBgn0265274 | Inx3 | 1.062 | 0.026 | - Innexin; gap junction protein; Involved in dorsal closure; intercellular transport; and phototransduction |
| FBgn0034259 | P32 | -1.001 | 0.022 | - Mitochondrial protein; Functions in presynaptic calcium signaling and neurotransmitter release; Chromatin metabolism |
| FBgn0033191 | CG1598 | -1.001 | 0.022 | - ATP binding; ATPase activity; Transport to ER |
| FBgn0031600 | CG3652 | -1.004 | 0.026 | - Unknown (Contains Yip1 domain) |
| FBgn0039371 | CG4960 | -1.018 | 0.022 | - Unknown (TB2/DP1/HVA22-related protein); Involved with regulation of intracellular transport |
| FBgn0033906 | ReepB | -1.018 | 0.022 | - Unknown (TB2/DP1/HVA22-related protein); Involved with ER organization and regulation of intracellular transport |
| FBgn0040602 | CG14545 | -1.054 | 0.022 | - Unknown |
| FBgn0034058 | Pex11 | -1.081 | 0.022 | - Unknown; Involved in peroxisome fission and organization |
| FBgn0010078 | RpL23 | -1.099 | 0.022 | - Myosin binding; Structural constituent of ribosome; Involved in translation |
| FBgn0013771 | Cyp6a9 | -1.161 | 0.022 | - Iron/heme binding; Involved in oxidation-reduction processes |
| FBgn0000615 | exu | -1.167 | 0.022 | - Single strand RNA binding; Protein homodimerization; Involved in embryonic pole axis specification, localization of bicoid and oskar mRNA |
| FBgn0086904 | Nacα | -1.238 | 0.022 | - Protein binding; Involved in neurogenesis, oogenesis, oskar mRNA localization |
| FBgn0051075 | CG31075 | -1.266 | 0.025 | - Aldehyde dehydrogenase (NAD) activity; Involved in metabolism and oxidation-reduction processes |
| FBgn0010038 | GstD2 | -0.993 | 0.022 | - Glutathione peroxidase activity, glutathione transferase activity |
| FBgn0001149 | GstD1 | -1.307 | 0.022 | - Glutathione transferase activity; DDT-dehydrochlorinase activity |
| FBgn0011761 | dhd | -1.348 | 0.022 | - Protein disulfide oxidoreductase activity (Thioredoxin domain); Involved in glycerol ether metabolism, cell redox homeostasis, and responses to DNA damage |
| FBgn0033979 | Cyp6a19 | -1.392 | 0.022 | - Cytochrome P450; electron carrier activity and heme binding; Involved in oxidation-reduction process |

** Knockdown (RNAi) lines not available.

differences we observe is striking given that only ~5–10% of the cell surface is wounded and undergoing repair.

We next determined if these genes were being co-differentially expressed by shared activating or regulatory elements within a localized region of the genome in response to wounding. Genome mapping of the 253 genes show no obvious clustering upon visual inspection (Fig 2D). Concomitantly, we mapped the 253 differentially-expressed genes onto the 1169 unique topologically associated domains (TADs) previously characterized in flies [45], and found no difference in overall differentially-expressed genes between TADs (p = 0.22), as well as when comparing just the down-regulated genes (p = 0.81) (Fig 2E). Interestingly, we detected a slight difference in differentially-expressed genes by TAD for up-regulated genes (p = 0.01), however

the majority of this signal appears to be driven by there being less up-regulated genes and many of these falling into TADs that were missing genes due to their poorer coverage on our arrays. The results from this TAD analysis suggest that the 253 genes are being regulated independently and deliberately in response to wound repair. Intriguingly, the 80 upregulated genes were, on-average, larger than gene products previously recorded during this stage of development (Fig 2F) [46–49], implicating the existence of a wound-repair specific program (see Discussion).

## Transcription is not required to initiate cell wound repair

We expected that if transcription served as an initiator for wound repair as previously proposed, then inhibition of transcriptional activity with α-amanitin, a transcription inhibitor that targets RNA polymerase thereby halting transcritional activity, would result in altered repair assessable by visualizing actin dynamics throughout the wound repair process. To ensure efficient transcriptional (or translational) knockdown, we verified the efficacy and duration of the α-amanitin, puromycin, or cycloheximide treatment using the MS2-MCP system, a visual reporter of active transcription (see Methods) (Fig 3A-3D') [50, 51]. In *Drosophila* syncytial embryos, GFP appears as puncta within the nuclei of control embryos indicative of active transcription, whereas these GFP puncta are absent in α-amanitin injected, but not puromycin or cycloheximide injected, embryos indicating that α-amanitin is effectively inhibiting transcription even beyond our initial wounding window (Fig 3A-3D'; S2 Video). To confirm our microarray results that transcription is unlikely to initiate repair in the *Drosophila* system, we then examined wound repair in nuclear cycle 4–6 *Drosophila* embryos that were injected with α-amanitin. Using time lapse microscopy and a fluorescent actin reporter, we find that in control embryos, where only buffer was injected, actin became enriched in two distinct locations: 1) adjacent to the wound edge, forming a robust actin ring, and 2) in a "halo" or diffuse accumulation along the outer periphery of the ring and identical to previous findings in uninjected embryos (Fig 3E-3E' and 3I-3K; S2 Video) [23, 52]. Consistent with our microarray results, α-amanitin injected embryos initially showed actin dynamics similar to those observed in control embryos, however they exhibited disruptions to the repair process during the subsequent actin remodeling phases (Fig 3F-3F' and 3I-3K; S2 Video). Thus, our results indicate that a transcriptional response is dispensable for the initiation of cell wound repair in the *Drosophila* model, but becomes important subsequently, potentially for replenishing and/or maintaining various factors necessary for establishing the wound repair response.

## The initial steps of cell wound repair are translation dependent

*Drosophila* early embryonic development is mostly driven by maternally deposited mRNA and protein until the maternal-to-zygotic genome transition (MZT) at nuclear cyle 14 (cf. [49]). To explore the role of translation in driving the wound repair process, embryos expressing a fluorescent actin reporter (sGMCA) were injected with the translation inhibitors puromycin (causes premature chain termination) or cycloheximide (blocks translational elongation) prior to laser wound induction (Fig 3C, 3D and 3G-3H'; S2 Video). While the wound fails to expand, some actin was recruited to the wound periphery, however, the actin ring/halo was not properly assembled and/or maintained resulting in aberrant spatiotemporal enrichment of actin (i.e. inside the wound area) (Fig 3G–3I; S2 Video). Quantitative measurements show a prolonged wound healing process compared to controls (Fig 3I), with significantly less wound expansion and slower wound closure (Fig 3J–3K). Taken together, our results suggest that the *Drosophila* embryo requires active translation to initiate wound repair, as well as to regulate actin dynamics throughout the repair process.

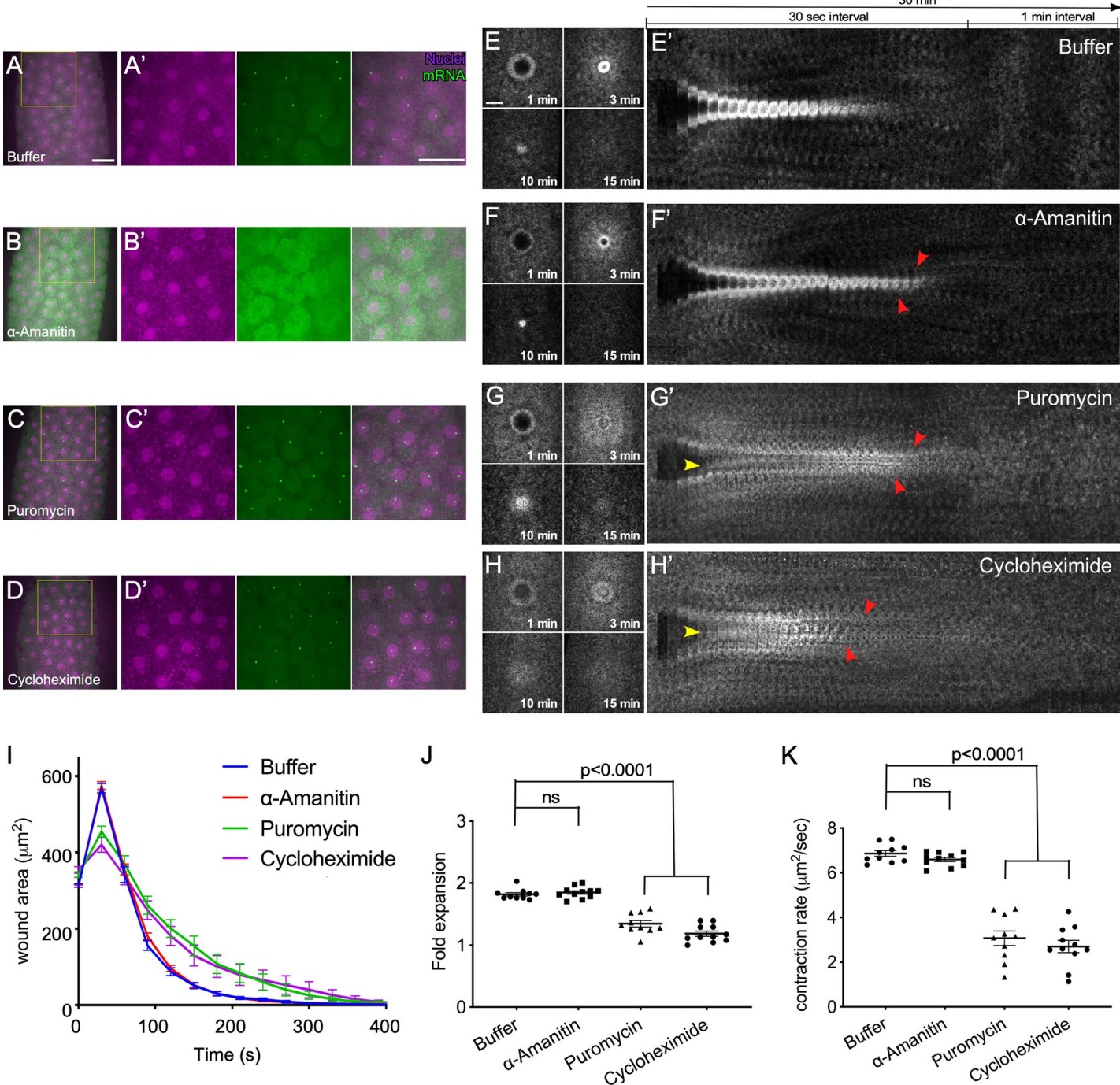

**Fig 3. Translation, rather than transcription, is needed for the initiation of cell wound repair.** (**A-D**) Confocal projections of a NC 10 embryo expressing the MS2-MCP system injected with: buffer (A), alpha-amanitin (B), puromycin (C), or cycloheximide (D). (**A'-D'**) higher magnification images of the respective regions in (A-D) demarcated by the yellow box, showing nuclei (magenta) and nascent mRNA (green). (**E-H**) Confocal projection stills from time-lapse imaging of actin dynamics (sGMCA) during cell wound repair in control (buffer only) (E), alpha-amanitin injected (F), puromycin injected (G), or cycloheximide injected (H) embryos. (**E'-H'**) XY kymographs across the wound areas depicted in E-H, respectively. Note extended actin remodeling (red arrowheads in b',c') and internal actin accumulation (yellow arrowhead in c'). (**I**) Quantification of wound area over time for (A-C'). Error bars represent ± SEM. (**J-K**) Quantification of wound expansion time (J) and wound closure speed (K) for conditions indicated. See S3 Table for numerical data. Student's t-test; all p-values indicated. Scale bars: 20 μm.

## Knockdown of differentially expressed genes results in wound over-expansion, abnormal actin dynamics, and remodeling defects upon wounding

We next examined the effects of removing the differentially-expressed genes on cell wound repair. We generated knockdown embryos for 15 of the top 16 up-regulated genes (Fig 4, Fig 5; S1 Fig, S2 Fig) and the 16 top down-regulated genes (Fig 5, Fig 6; S2Q Fig, S3 Fig) based on their fold-change (Table 1) by expressing RNAi constructs in the female germline using the GAL4-UAS system [53, 54]. We then observed actin dynamics following laser wounding using a fluorescent actin reporter (sGMCA). In all 31 cases, the wounded knockdown embryos exhibited disruptions at various post-initiation steps of the cell wound repair process, including wound over-expansion (Fig 5A and 5E), delayed/altered rates of wound contraction (Fig 5B and 5F), aberrant actin dynamics (Fig 5C, 5D, 5G and 5H), and/or remodeling defects (Fig 4, Fig 6; S2 Fig, S3 Fig). Examples of these phenotypes are described below.

**Up-regulated genes.** The *Drosophila* embryo is under tension such that when it is wounded, the plasma membrane and cortical actin cytoskeleton recoil slightly leading to an expansion of the wound [23, 55]. Interestingly, wounds generated in knockdowns of three of the up-regulated genes (*Inx3, CG43963, danr*) failed to expand, whereas others (*Dtg, link, l(3) neo38, Egfr, dpn*) exhibited wound over-expansion (Fig 4, Fig 5A; S2 Fig). Similarly, wounds generated in knockdowns of three of the up-regulated genes (*Inx3, ImpL2, link*) exhibited slower wound contraction rates, whereas others (*l(3)neo38, CG43963, Egfr, Ama*) exhibited faster wound contraction rates compared to control wounds (Fig 4, Fig 5B; S2 Fig).

In all 15 cases of RNAi knockdown for up-regulated genes, wounded embryos exhibited abnormal actin dynamics, including premature actin ring/halo disassembly, failure of actin ring/halo dissassembly, and/or abnormal actin ring/halo disassembly with concomitant accumulation of actin within the wound. (Fig 4, Fig 5C and 5D; S3 Video; S2 Fig). Wounds generated in knockdowns of *Imaginal morphogenesis protein-Late 2* (*ImpL2*) and *Epidermal growth factor receptor* (*Egfr*) are exemplified by their incomplete formation and premature dissassembly of the actomyosin ring causing rifts at the initial injury site that remained open for the entire time of repair (Fig 4B–4C', 4J and 4K; S3 Video). ImpL2 has been proposed to work antagonisitically to the insulin/insulin-like (IIS) signaling pathway by interacting with receptor/ligand interactions to inhibit downstream signal transduction [56–58]. Egfr encodes a receptor tyrosine kinase that works upstream of the c-jun N-terminal kinase (JNK) and decapentaplegic (dpp) pathways. Loss of Egfr results in down-regulation of JNK activity leading to the impairment of dorsal closure, a process sharing many features with epithelial (multicellular) wound repair [59]. Wounds generated in knockdowns of *jitterbug* (*jbug*) and *nullo*, are characteristically defined by the pronounced formation of actin inside the wound area (Fig 4G–4H' and 4O–4P; S4 Video). Jbug is a filamin-type protein that serves as an F-actin crosslinker providing stability to the cytoskeleton, a system that has been proposed to utilize mechanical cues such as tension to modulate cellular processes [60, 61]. Nullo has been shown to establish cortical compartments during cellularization of the *Drosophila* embryo, suggesting an important role regulating actin stability at the cortex [62, 63].

Following wound closure, extensive remodeling of the cortical cytoskeleton and its overlying plasma membrane is necessary to re-establish normal architectures and activities. Wounds generated in knockdowns of Gp150, Inx3, and Thor, are unable to resolve actin structures and/or properly remodel cortical actin after wound closure (Fig 4D–4F' and 4L–4N; S3 Video). Gp150 encodes a transmembrane glycoprotein that regulates Notch signaling during normal eye development in *Drosophila* [64], whereas Inx3 encodes a gap junction protein involved in morphogenesis and nervous system development [65, 66]. Thor encodes a

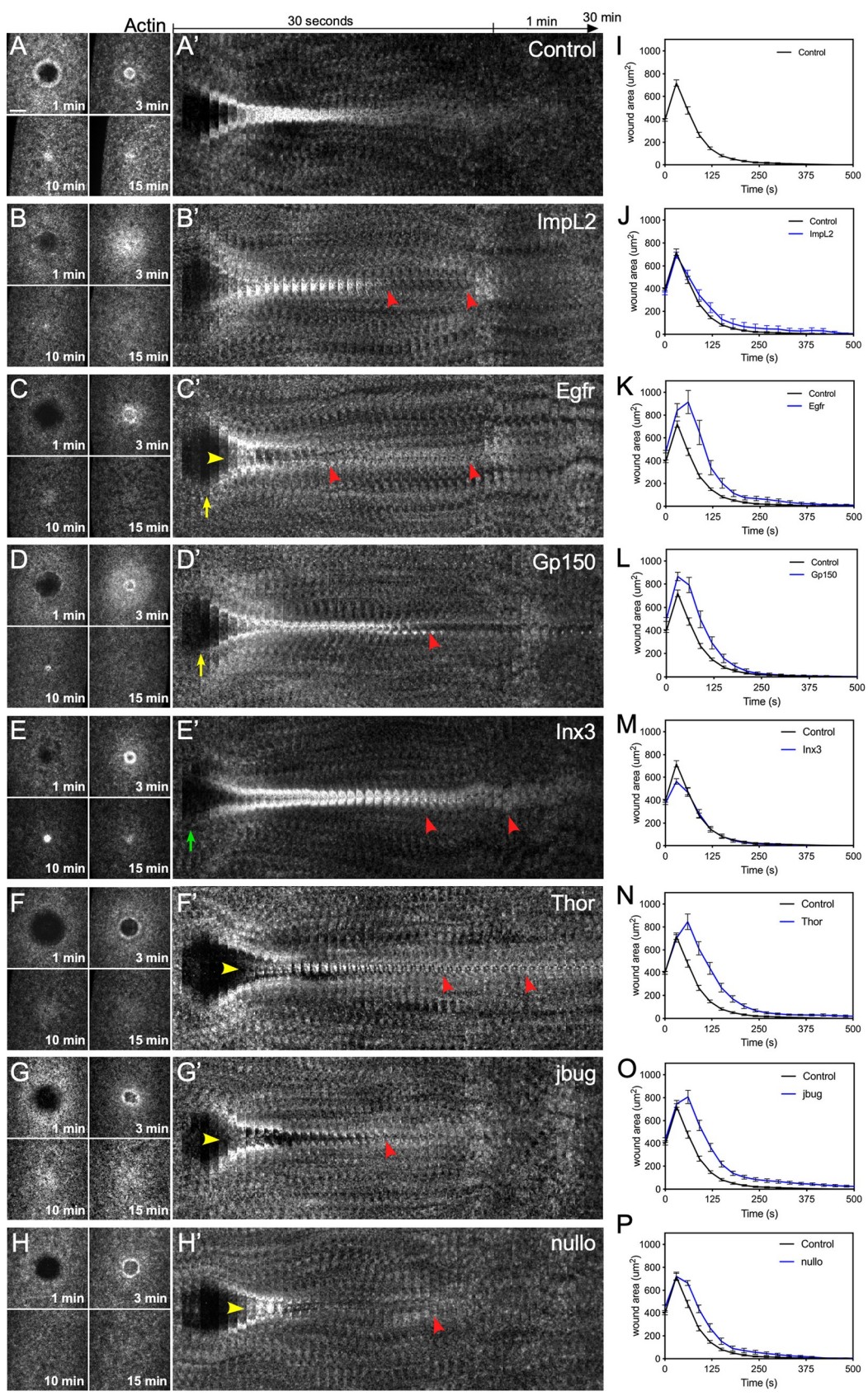

**Fig 4. Knockdown of up-regulated genes results in wound over-expansion and abnormal actin dynamics.** (**A-H**) Confocal XY projections of actin dynamics at 1, 3, 10, and 15 mpw from *Drosophila* NC4-6 embryos coexpressing sGMCA and a UAS-RNAi transgene during cell wound repair for control (w[1118]/+; sGMCA, 7063/+) (A), ImpL2[RNAi]/+; sGMCA, 7063/+ (B), EGFR[RNAi]/+; sGMCA, 7063/+ (C), Gp150[RNAi]/sGMCA, 7063 (D), Inx3[RNAi]/sGMCA, 7063 (E), Thor[RNAi]/ sGMCA, 7063 (F), Jbug[RNAi]/sGMCA, 7063 (G), Nullo[RNAi]/sGMCA, 7063 (H). (A'-H') XY kymographs across the wound areas depicted in (A-H), respectively. Note wound overexpansion (yellow arrows), wound underexpansion (green arrows), internal actin accumulation (yellow arrowhead), and remodeling defect/open wound (red arrowhead). (**I-P**) Quantification of wound area over time for (A-H'), respectively. Error bars represent ± SEM; n ≥ 10. See S3 Table for numerical data. Scale bars: 20 μm.

translation inhibitor functioning downstream of insulin signaling that is sensitive to reactive oxygen species [67]. Interestingly, like ImpL2, Thor is a IIS pathway constituent and Gp150 Down-regulated geneshas also been shown to physically interact with components of this pathway (Pten and S6k) [68].

**Down-regulated genes.** Interestingly, in all 16 cases of RNAi knockdown for the down-regulated genes examined, wounded embryos exhibited abnormal cell wound repair dynamics that included the same major, but non-mutually exclusive, steps as described above for the up-regulated genes. A number of the genes that were downregulated have an unknown molecular function and/or associated biological processes (Table 1; Fig 5E–5H, Fig 6; S4 Video; S2Q Fig, S3 Fig). Of these unknown genes, CG31075 underwent a mild expansion followed by a contraction rate similar to that in wildtype, albeit with incomplete wound closure (Fig 5E, Fig 6A–6A' and 6H; S4 Video), CG4960 exhibited a slight delay in wound repair dynamics but retained noticeably enriched actin structures after closure (Fig 6E–6E' and 6L; S4 Video), and CG1598 developed a visually distinct, but transient, enrichment of actin inside the wound area prior to closure (Fig 6G–6G' and 6N; S4 Video). Of genes with known motifs/functions, Glutatione S transferases D2 (GstD2) and D1 (GstD1) RNAi knockdowns showed similar phenotypes exhibiting a short-lived accumulation of actin inside the wound area and delayed closure dynamics during the initial steps of repair (Fig 6B–6B', 6F–6F', 6I and 6M; S4 Video), and in later steps, both are unable to completely close (Fig 6B–6B' and 6I; S4 Video). Wound repair begins normally in *exu* knockdowns, however the leading edge and surrounding actin structures soon become static resulting in an open wound area and prolonged actin accumulation (Fig 6D–6D' and 6K; S4 Video). In addition to the phenotypes described above, many of these knockdowns exhibit wound over-expansion (*CG3652*, *P32*, *dhd*, *RpL23*, *Cyp6a9*, *Cyp6a19*) (Fig 5E, Fig 6; S4 Video; S3 Fig) and nearly all exhibit remodeling defects (Fig 6, red arrowheads; S4 Video; S3 Fig, red arrowheads). Thus, in all 31 cases of up- or down- regulated genes examined, knockdown using RNAi transgenes resulted in abnormal cell wound repair. Despite the molecular functions of many of these genes being unknown, they have been implicated in various cellular processes, but most notably a subset are involved in insulin signaling.

## Activation of insulin/insulin-like (IIS) constituents during normal wound repair

Deficiencies in insulin signaling have been implicated in multicellular (tissue/epithelial) repair, where it is thought to impede growth factor production, angiogenic response, and epidermal barrier function [69–72], functions that might not normally be expected to govern regulation within individual cells. Deficiencies in insulin signaling have also been associated with diabetic myocytes that exhibit defects in cell wound repair [17]. While the diabetic environment is multifaceted and the repair deficiencies observed could be indirect effects, prolonged exposure of cultured C2C12 skeletal muscle myocytes to high glucose levels is sufficient to induce the repair defect, suggesting direct participation of insulin signaling [17]. The fact that ImpL2 and Thor, two of the most upregulated genes in our analyses, are constituents of the insulin/

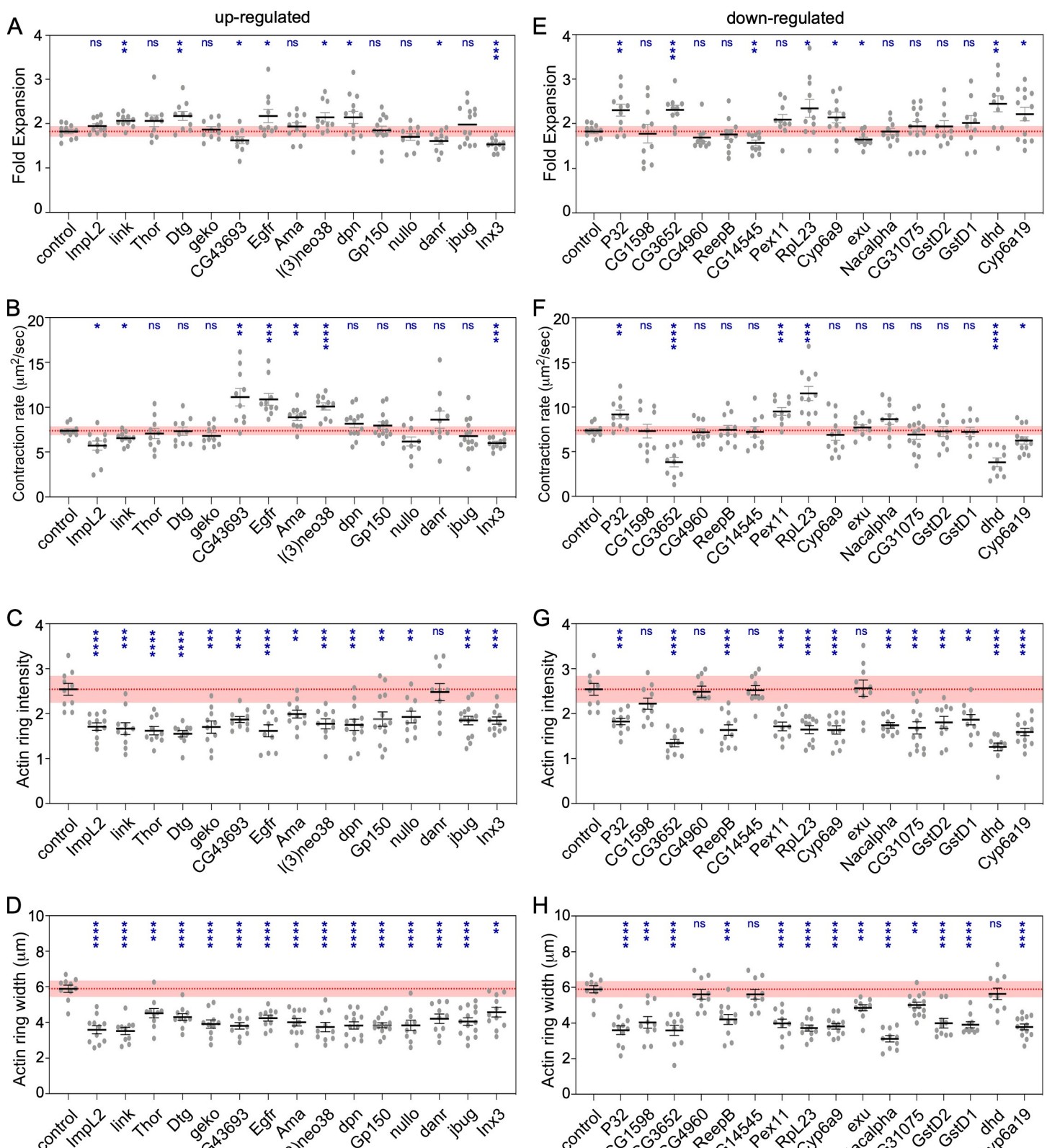

**Fig 5. Quantification of wound and actin dynamics in control and knockdowns for upregulated and downregulated genes.** (**A-D**) Quantification of wound expansion (A), contraction rate (B), actin ring intensity (C), and actin ring width (D) from control (sGMCA, 7063/+) and knockdowns for all 15 up-regulated genes (RNAi/+; sGMCA, 7063/+ or sGMCA, 7063/RNAi). (**E-H**) Quantification of wound expansion (E), contraction rate (F), actin ring intensity (G), and actin ring width (H) from

control (sGMCA, 7063/+) and knockdowns for all 16 down-regulated genes (RNAi/+; sGMCA, 7063/+ or sGMCA, 7063/RNAi). Black line and error bars represent mean ± SEM. Red line and square represent mean ± 95% CI from control. n ≥ 10. See S3 Table for numerical data. Student's t-test is performed to compare control with knockdowns. * is p<0.05, ** is p<0.01, *** is p<0.001, **** is p<0.0001, and ns is not significant.

insulin-like growth factor signaling (IIS) pathway in *Drosophila* (Fig 7A) was still somewhat unexpected.

PIP$_3$ (phosphatidylinositol (3,4,5)-triphosphate) is a phospholipid that composes a subset of specialized plasma membrane with various trafficking and signaling related functions [73], including recruitment to cell wounds [74]. A PIP$_3$-GFP reporter construct has also been shown to function as a reporter of insulin signaling activity [75]. To determine if the canonical IIS pathway was involved in individual cell wound repair, we first examined the recruitment pattern of a PIP$_3$, co-expressed with a Cherry fluorescently-tagged actin reporter (sChMCA), in a wildtype and *chico* RNAi knockdown background (Fig 7B–7F). PIP$_3$-GFP is recruited to same region as the actomyosin ring in wildtype embryos (Fig 7B-B", 7D and 7F), confirming the requirement for autocrine insulin pathway signaling. Importantly, this recruitment is dependent on the upstream activation of the insulin receptor (InR), as PIP$_3$-GFP recruitment is disrupted in a *chico* RNAi background (Fig 7C–7C", 7E and 7F).

We next examined the wound repair phenotypes in knockdown backgrounds for components spanning the IIS pathway by expressing RNAi constructs for pathway components in the female germline using the GAL4-UAS system [53, 54], then observing actin dynamics using a fluorescent actin reporter (sGMCA). The one ligand and six of the major IIS pathway components tested—Ilp4 (Insulin-like peptide), InR (Insulin receptor), Chico (IRS homolog), Pi3K21B (Phosphoinositide3-Kinase), Akt1 (Kinase), FoxO (transcription factor), and Reptor (transcription factor)—exhibited abherrant wound repair with overlapping phenotypes reflecting involvement at several steps in the repair process (Fig 7A, 7G–7J, Fig 8; S5 Video; S2Q Fig). With the exception of ImpL2, Ilp4, and InR (components at the top of the pathway), mutants for IIS pathway components exhibited wound overexpansion immediately after laser ablation that was visible as the outward retraction of the wound edge (Fig 7G and 7H, Fig 8; S5 Video). Following this overexpansion, actin structures became transiently enriched inside the wound area, but dissassembled prior to complete wound closure (Fig 7I and 7J, Fig 8; S5 Video). Lastly, progression of wound closure was signficantly delayed and/or incomplete, leaving openings around the actin ring as it translocated (Fig 8, red arrowheads; S5 Video). While we can not rule out contributions from non-canonical insulin signaling pathways, our results show that key components of the canonical insulin signaling pathway are not only called to a wound, but have detrimental effects on actin and wound dynamics upon knockdown. Collectively, our results suggest that a tight association exists between the factors that regulate both insulin signaling and cell wound repair in the *Drosophila* model.

## The IIS pathway effectors Profilin (Chickadee) and Girdin are required for cell wound repair

The IIS pathway has recently been shown to control actin dynamics independently of its role in growth control [76]. In particular, the IIS pathway has been found to activate the expression of the *Drosophila* profilin homolog (*Chickadee*), as well as the Akt substrate Girdin (GIRDers of actIN; also known as GIV) [76–78]. To determine if these actin regulators function as IIS pathway effectors during cell wound repair, we stained wounded embryos that expressed a GFP-tagged actin reporter (sGMCA) in a wildtype or *chico* RNAi knockdown background with antibodies to Profilin/Chickadee and Girdin (Fig 9A–9D). Both proteins are recruited to wounds, although their spatial recruitment patterns are not the same. Girdin exhibits a

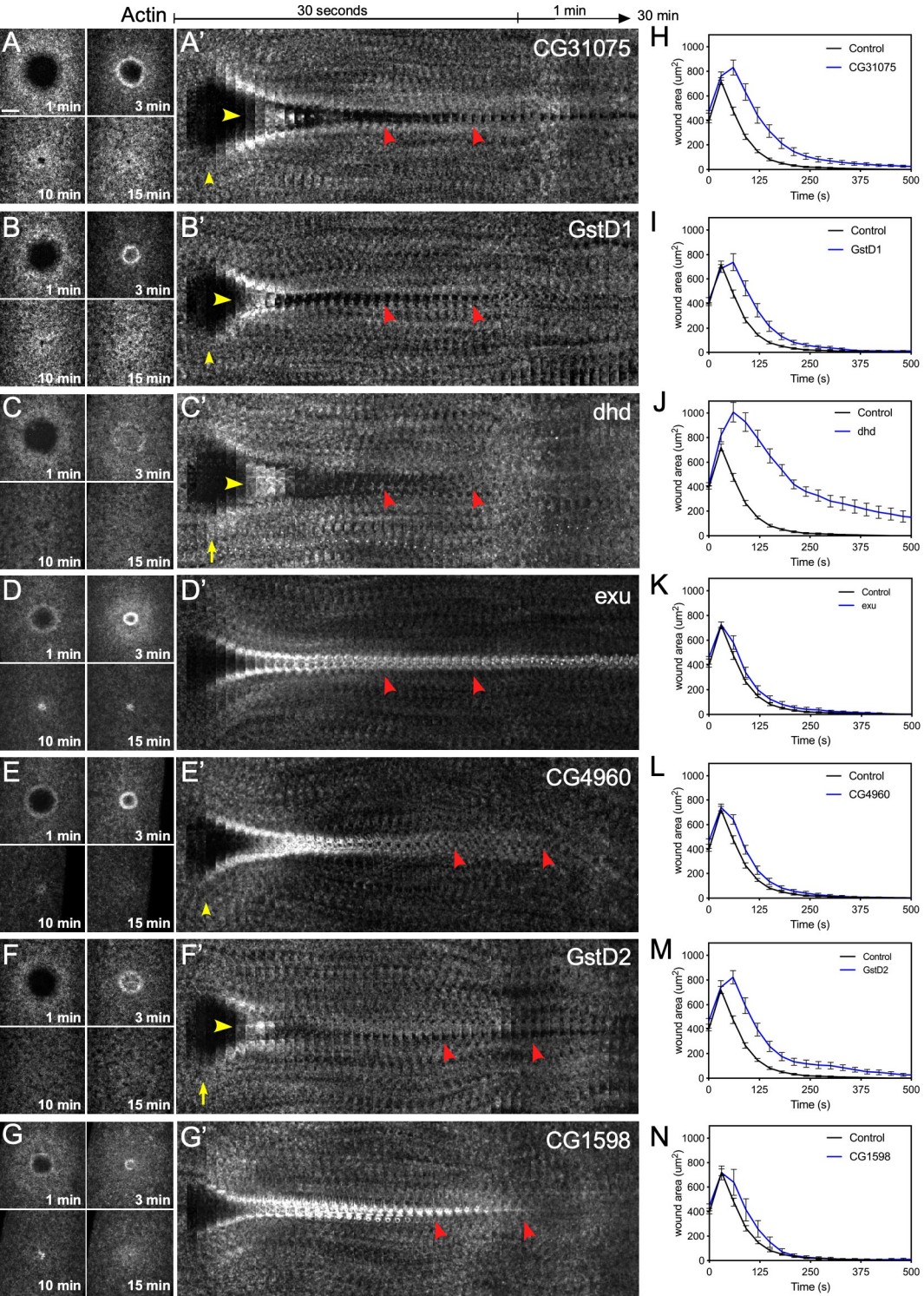

**Fig 6. Knockdown of down-regulated genes results in wound over-expansion and abnormal actin dynamics. (A-G)** Confocal XY projections of actin dynamics at 1, 3, 10, and 15 mpw from *Drosophila* NC4-6 embryos coexpressing sGMCA and a UAS-RNAi transgene during cell wound repair for CG31075[RNAi]/+; sGMCA, 7063/+ (A), GstD1[RNAi]/+; sGMCA, 7063/ + (B), dhd[RNAi]/+; sGMCA, 7063/+ (C), Exu[RNAi]/+; sGMCA, 7063/+ (D), CG4960[RNAi]/+; sGMCA, 7063/+ (E), GstD2[RNAi]/+; sGMCA, 7063/+ (F), CG1598[RNAi]/+; sGMCA, 7063/+ (G). **(A'-G')** XY kymographs across the wound areas depicted in (A-G), respectively. Note wound overexpansion (yellow arrows), wound underexpansion (green arrows), internal actin accumulation (yellow arrowhead), and remodeling defect/open wound (red arrowhead). **(H-N)** Quantification of wound area over time for (A-G'), respectively. Error bars represent ± SEM; n ≥ 10. See S3 Table for numerical data. Scale bars: 20 μm.

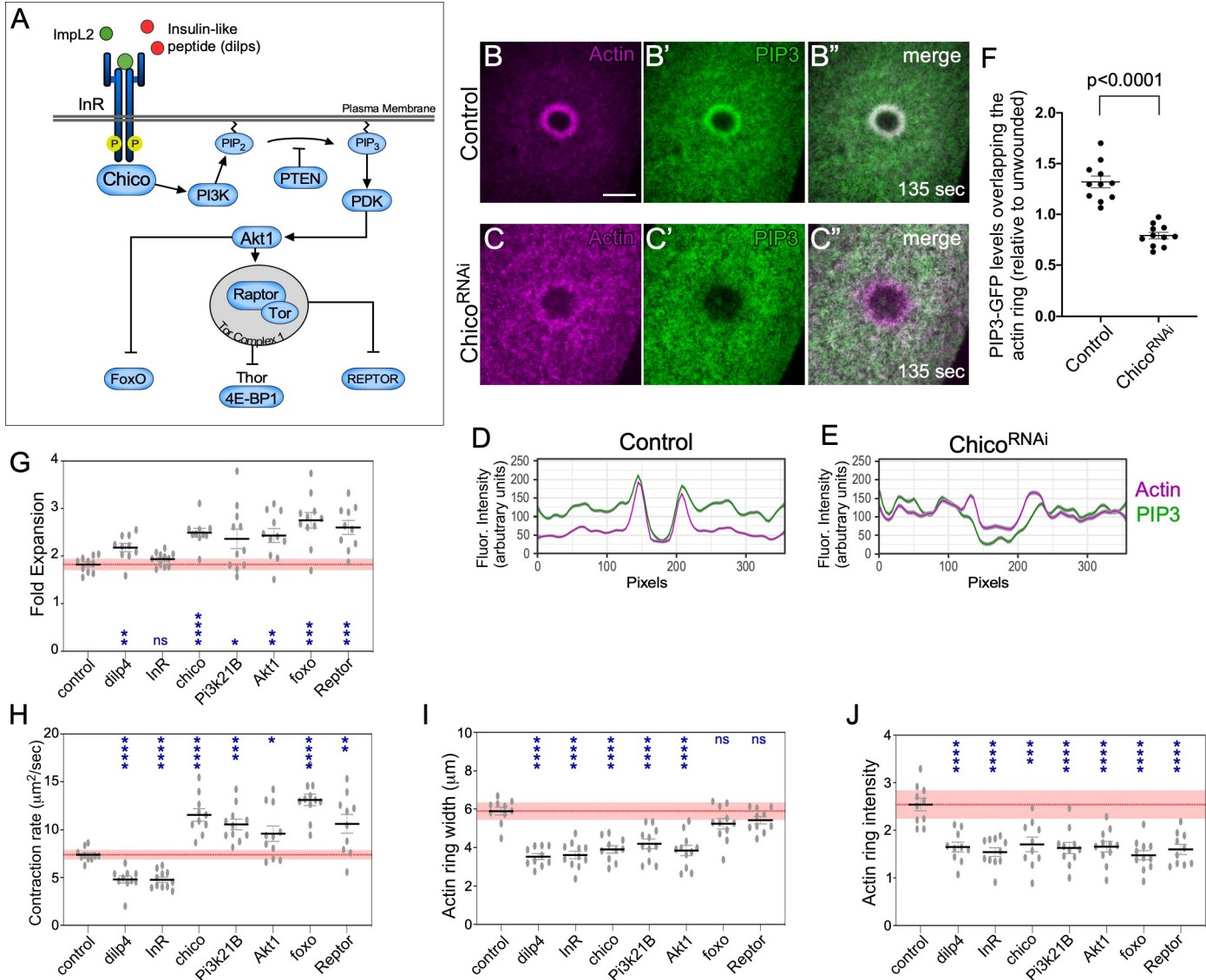

**Fig 7. Localization of IIS pathway components.** (**A**) Simplified diagram of the IIS pathway in *Drosophila* showing the components tested using GFP reporters and RNAi transgenes. (**B-C"**) Confocal xy projection images from *Drosophila* NC4-6 staged embryos co-expressing an actin marker (sChMCA) and GFP-tagged PIP3 in a control (B-B") or chico RNAi (C-C"). (**D-E**) Smoothened fluorescence intensity (arbitrary units) profiles derived from averaged fluorescence intensity values over a 10 pixel width across the wound area in the embryo shown (B-C"), respectively. Gray area represents the 95% CI. Scale bars: 20 μm. (**G-J**) Quantification of wound expansion (G), contraction rate (H), actin ring intensity (I), and actin ring width (J) from control (sGMCA, 7063/+) and knockdowns for IIS pathway genes (RNAi/ +; sGMCA, 7063/+ or sGMCA, 7063/RNAi). Black line and error bars represent mean ± SEM. Red line and square represent mean ± 95% CI from control. n ≥ 10. See S3 Table for numerical data. Student's t-test is performed to compare control with knockdowns. * is p<0.05, ** is p<0.01, *** is p<0.001, **** is p<0.0001, and ns is not significant.

punctate recuitment at wounds with the highest accumulation overlapping the membrane plug inside the actin ring and with lower level diffuse accumulation overlapping the actin ring and the innermost part of the actin halo (Fig 9A and 9B). Profilin/Chickadee recruitment is internal to the actin ring and appears to be excluded from the actin ring region (Fig 9A and 9B). Importantly, the accumulation of both Profilin/Chickadee and Girdin at wounds requires a functioning IIS pathway as these accumulations are lost in a *chico* RNAi background (Fig 9C and 9D).

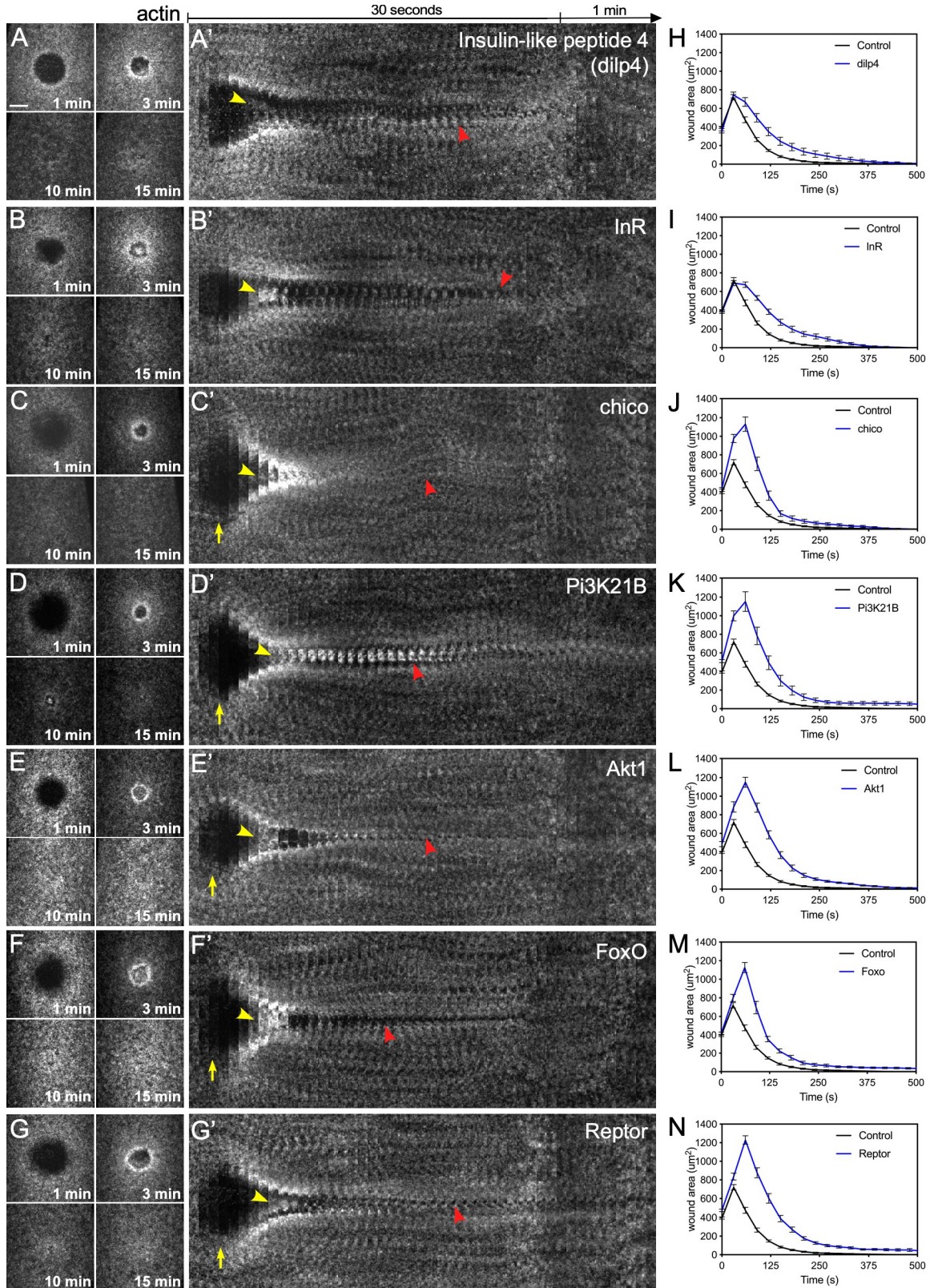

**Fig 8. Actin dynamics of insulin/insulin-like (IIS) pathway mutants.** (**A-G**) Confocal XY projections of actin dynamics at 1, 3, 10, and 15 mpw during cell wound repair in *Drosophila* NC4-6 embryos expressing sGMCA and a mutant for insulin-like peptide 4 (*Ilp4[1]*; A),

or a UAS-RNAi transgene for InR$^{RNAi(1)}$/+; InR$^{RNAi(2)}$/sGMCA, 7063 (B), Chico$^{RNAi}$/sGMCA, 7063 (C), Pi3K21B$^{RNAi}$/sGMCA, 7063 (D), Akt1$^{RNAi}$/sGMCA, 7063 (E), FoxO$^{RNAi}$/sGMCA, 7063 (F), and Reptor$^{RNAi}$/sGMCA, 7063 (G). (**A'-G'**) XY kymographs across the wound areas depicted in (A-G), respectively. Note wound overexpansion (yellow arrows), wound underexpansion (green arrows), internal actin accumulation (yellow arrowhead), and remodeling defect/open wound (red arrowhead). (**H-N**) Quantification of wound area over time for (A-G'), respectively. Error bars represent ± SEM; n ≥ 10. See S3 Table for numerical data. Scale bars: 20 μm.

We next examined the effects of removing Girdin and Profilin/Chickadee on cell wound repair. Similar to knockdown of IIS pathway components described above, Girdin RNAi knockdown embryos exhibited aberrant wound repair including wound overexpansion, enrichment of actin structures inside the wound area, and signficantly delayed wound closure (Fig 9E–9E', 9G and 9I–9L; S5 Video; S2Q Fig). Unfortunately, Profilin/Chickadee RNAi knockdown females do not produce eggs. We therefore used the *wimp* mutation [79, 80] to generate reduced Profilin/Chickadee expression in both the germline and soma (*wimp* reduces maternal gene expression such that, when *in trans* to the *chickadee*$^{221}$ allele, it effectively generates a strong *chickadee* hypomorph, referred to as reduced Profilin). Similar to knockdown of Girdin and IIS pathway components, reduced Profilin/Chickadee embryos exhibited wound overexpansion, enrichment of actin structures inside the wound area, and signficantly delayed wound closure (Fig 9F–9F', 9H and 9I–9L; S5 Video; S2Q Fig). Thus, our results indicate that Girdin and Profilin/Chickadee are actin regulatory downstream effectors of the IIS pathway in cell wound repair.

## Discussion

Our study shows that cellular wound repair is not dependent on transcriptional activity to initiate wound repair programs, that dormant transcription pathways are activated in response to wounds, and that the insulin signaling pathway is an essential component of the repair process. A calcium influx-triggered transcriptional response was thought to be important to lead off the cell wound repair process, eliciting a downstream wound repair program. However, this proposed mechanism was at odds with the *Drosophila* syncytial embryo cell wound model that faithfully recapitulates the majority of features associated with other single cell wound repair models (*Xenopus* oocytes, tissue culture cells, sea urchin eggs) [2, 3, 18, 23, 27, 52, 81–83], yet represents a special system running mostly off of maternally contributed products, highlighted by rapid cell cycles (~10 minutes/cycle) and minimal zygotic transcription [48, 49, 84].

Consistent with the closed nature of the *Drosophila* syncytial embryo cell wound model, we find no altered gene expression immediately upon wounding either as assayed by microarray analysis of laser wounded versus unwounded embryos or following injection of the α-amanitin transcriptional inhibitor. However, we find that injection of translational inhibitors prior to wounding leads to severe wound repair defects, indicating that the initial steps of cell wound repair require protein synthesis. This was surprising given that the initial steps of wound repair are extremely rapid (i.e., the hole is plugged and an actomyosin ring forms within 30–60 seconds). However, there is precedence for local translation of molecules such as Rho GTPase, GAP-43, and CYFIP1, allowing rapid regulation of specific cellular and developmental events, including growth cone elongation or collapse and dendritic spine formation [85–88].

We do detect alterations in gene expression at subsequent stages in the repair process: we identified 253 genes (out of ~8000 genes assayed) whose expression is significantly up (80 genes) or significantly down (173 genes) following laser wounding. Polymerase rates in the early *Drosophila* embryo were reported to be 1.1–1.5 kb/min, leading to the suggestion that any genes transcribed in the early *Drosophila* embryo prior to the mid-blastula transition must be small with minimal introns due to the rapid (~10 min) cell cycles and limited transcription

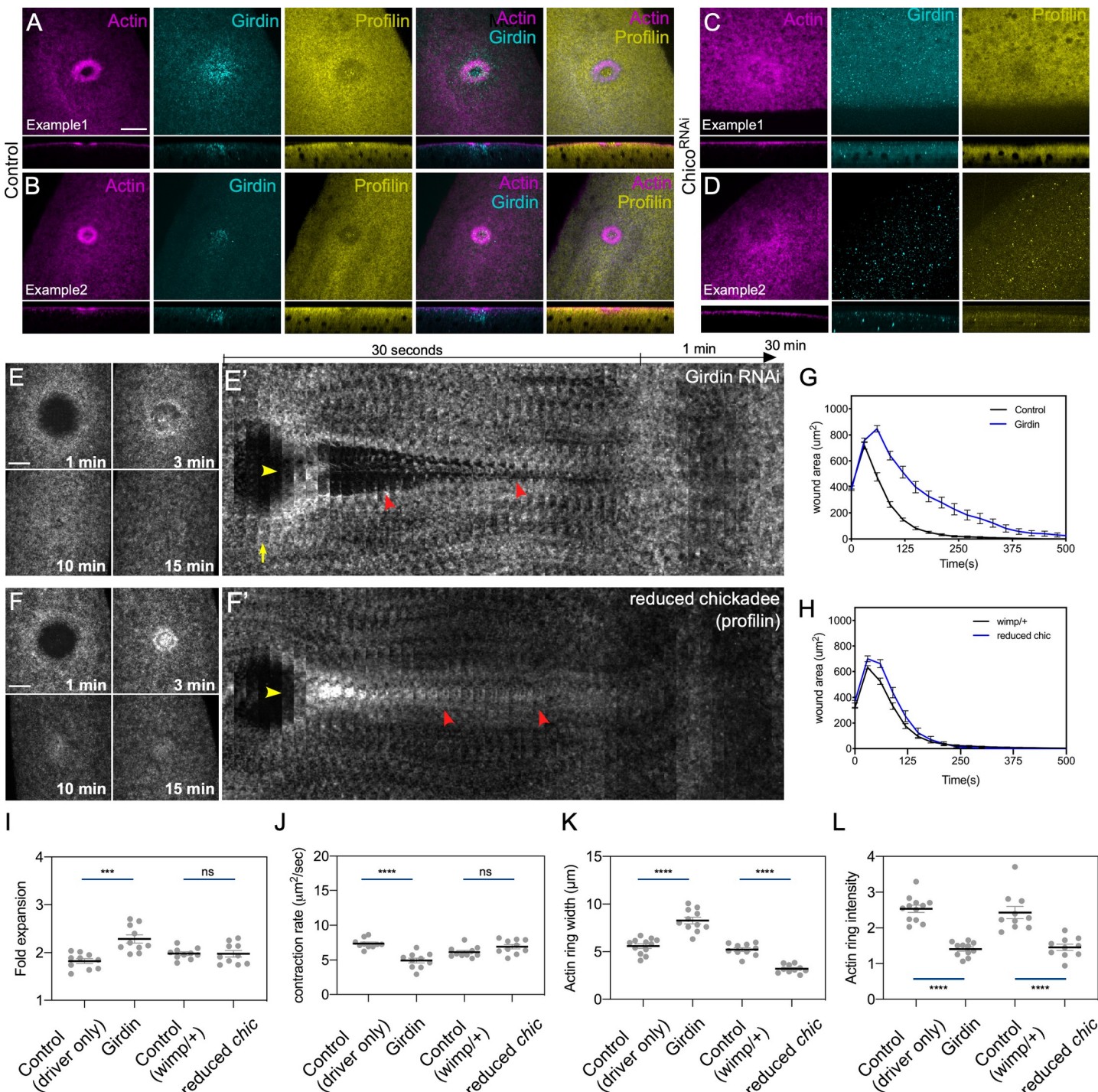

**Fig 9. Chickadee (profilin) and Girdin are insulin/insulin-like (IIS) pathway effectors during cell wound repair.** (**A-D**) Confocal XY projections of laser wounded *Drosophila* NC4-6 wildtype (A-B) or *chico* RNAi knockdown (C-D) embryos stained for Girdin (Girdin), Chickadee/profilin (Profilin), and F-actin/phalloidin (Actin). (**E-F'**) Confocal XY projections of actin dynamics at 1, 3, 10, and 15 mpw from *Drosophila* NC4-6 embryos coexpressing sGMCA and a UAS-RNAi transgene during cell wound repair for Girdin$^{RNAi}$ (Girdin$^{RNAi}$/+; sGMCA, 7063/+) (E) and reduced chickadee (sGMCA; *chickadee$^{221}$*/+ sGMCA, *wimp*/+) (F). (E'-F') XY kymographs across the wound areas depicted in E-F, respectively. Note wound overexpansion (yellow arrows), wound underexpansion (green arrows), internal actin accumulation (yellow arrowhead), and remodeling defect/open wound (red arrowhead). (**G-H**) Quantification of wound area over time for (E-F'), respectively. Error bars represent ± SEM; n ≥ 10. (**I-L**) Quantification of wound expansion (G), contraction rate (H), actin ring intensity (I), and actin ring width (J) from control (sGMCA, 7063/+) and knockdowns for IIS pathway genes (RNAi/+; sGMCA, 7063/+ or sGMCA, 7063/RNAi). Error bars represent ± SEM; n ≥ 10. See S3 Table for numerical data. Student's t-test is performed to compare control with knockdowns. *** is p<0.001, **** is p<0.0001, and ns is not significant. Scale bars: 20 μm.

time [48, 49, 89–95]. Recent studies have revised this rate to 2.4–3.0 kb/min, lowering the size constraints on the zygotic genes that can be successfully transcribed prior to the mid-blastula transition[47]. Therefore, genes up to ~20–25 kb could theoretically be transcribed during the early and rapid *Drosophila* embryo cell cycles. In this case however, the number of mRNA molecules would be likely limited by the lower number of nuclei present and thus copies of DNA.

We find that the average size of transcripts in syncytial *Drosophila* embryos is 2.5 kb, similar to the previously reported size of 2.2 kb (compared to the overall average length of coding genes in *Drosophila* of 6.1 kb) [46, 96]. Genes whose expression goes down during wound repair are, on average, 1.9 kb. It is intriguing that these actively down-regulated genes negatively impact the wound repair process when knocked-down. These genes likely represent RNAs stored in the embryo that are used up during the repair process and not replaced. Alternatively, it is possible that wound repair itself may slightly delay development leading to a subset of zygotically expressed genes whose expression is lagging behind in wounded versus unwounded embryos such that this delayed developmental upregulation is read out as a down-regulation of genes.

Genes whose expression goes up during wound repair are likely to include those encoding cellular components that were expended during the repair process and are being replenished for normal developmental events to proceed. Surprisingly, we find that the up-regulated genes are much larger on average (3.7 kb) than the average sized transcript at that stage (2.5 kb). Given the rapid mitotic divisions (~10 min) and fixed transcription rate during the syncytial embryo stage, it was proposed that long genes are either not transcribed, transcription is aborted, or long transcripts are subject to specific developmental programs that truncate them to allow subsets of their functions [48, 94, 97]. Thus, these larger genes also likely include cellular components that are activated specifically for the repair process. Indeed, this subset of "up-regulated" genes includes genes that are not usually expressed in the syncytial *Drosophila* embryo (e.g., CG43693). Thus, our results suggest that, when wounded, the embryo may be able to activate a transcriptional program that is usually dormant during these stages.

Interestingly, 2 of the top 3 genes whose expression is significantly higher following wounding—ImpL2 and Thor—are components of the Insulin signaling pathway. While it has been shown that defective insulin signaling impairs epithelial (multicellular) wound repair [69–72], our findings lend support to the emerging idea that insulin signaling is also directly required for wound repair within single cells [17, 74]. Using a combination of RNAi knockdowns and GFP reporters, we have shown that all major components of the IIS pathway are involved in cellular wound repair, and upon knockdown, display similar phenotypes, suggesting that in this context the canonical IIS pathway activation occurs in an autocrine-like manner. Previous studies have highlighted the necessity of calcium influx to facilitate vesicle exocytosis and subsequent fusion of the plasma membrane during wound repair [18, 19, 81, 82]. Similarly, this influx has also been shown to modulate insulin secretion in *β*-islet cells via the opening of L-type channels by establishing calcium microdomains along the cortex [98, 99]. Insulin/insulin-like peptides are secreted into the extracellular space where they bind to InR thereby activating the heavily conserved IIS pathway that is known to regulate a number of downstream processes that range from transcription via phosphorylation events on the FOXO family of transcription factors to translation via the regulation of the 4E-binding protein, Thor [71, 100–103]. Recently emerging evidence has also shown that the activated IIS pathway can control actin dynamics through activation of actin regulators including Chickadee (profilin) and Girdin [76–78, 104, 105].

Observation of actin dynamics in mutants for a number of the IIS pathway components show common phenotypes of impaired cytoskeleton dynamics, most notably an immediate

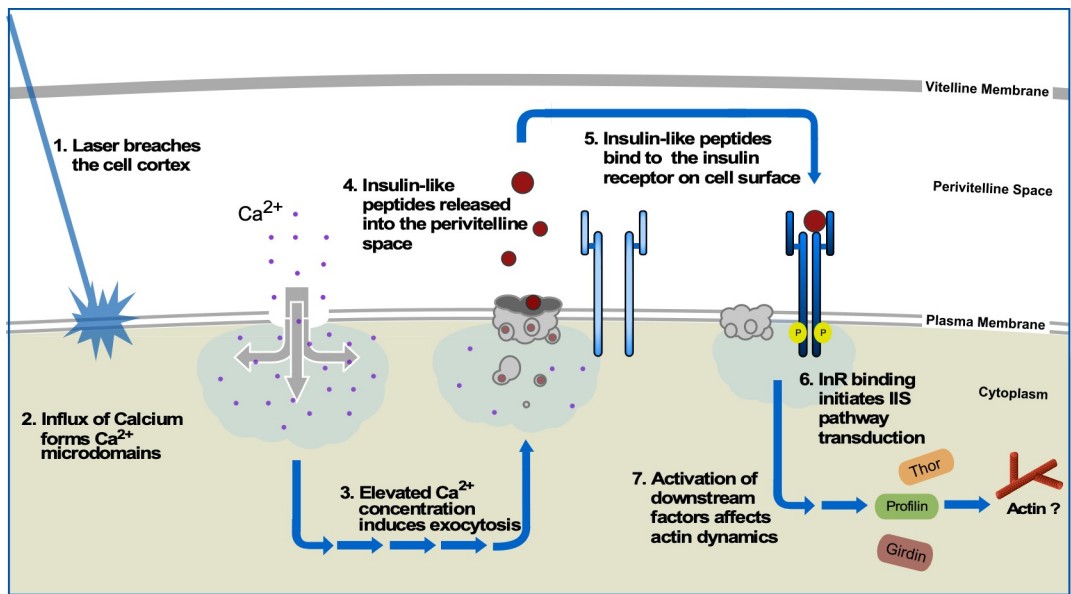

**Fig 10. Model for insulin/insulin-like (IIS) pathway function in cell wound repair.** Wounding of the cell cortex leads an rush of calcium into the cytoplasm developing locales of increased calcium concentration known as microdomains. To plug the hole, elevated calcium levels initiate exocytotic programs to recruit vescles to the wound area, forming a plug while simultaneously releasing insulin-like peptide 104 (ilp4) into the perivitelline space. Ilp4 is recognized by the insulin receptor (InR) where it binds and activates the IIS pathway. Subsequent phosphorylation events downstream of InR, ultimately activate downstream effectors including the actin remodelers Chickadee and Girdin to repair and restore the cortex back to its normal state.

over-expansion of the wound leading edge and a transient actin structure forming inside the wound area suggesting that normal wound repair processes are heavily reliant on a functioning IIS pathway. We propose that the initial inrush of calcium generates microdomains that trigger the secretion of the *Drosophila* insulin-like peptide 4 (Ilp4) into the perivitelline space where it recognizes and binds to the extracellular face of the Insulin receptor (Fig 10, steps 1–4). Subsequently, the InR is activated and initiates a signaling cascade that regulates a number of downstream processes, including cytoskeletal dynamics (Fig 10, steps 5–7). Chickadee/profilin binds to actin and affects the formation/remodeling of actin-rich structures [76]. Girdin also binds to actin, as well as the catenin-cadherin complex and the Exo-70 subunit of the exocyst complex, where it has been proposed to coordinate cytoskeleton organization, cell adhesion, membrane trafficking events, and serves as an indicator for poor prognosis with invasive breast cancers [77, 78, 104, 106, 107]. Interestingly, *girdin* and Profilin knockdown embryos exhibit wound repair phenotypes consistent with defects in actin structure assembly/remodeling, actomyosin ring attachment to the overlying plasma membrane, and membrane trafficking. In addition to the genes involved in the IIS pathway, our microarray analyses identified numerous other genes that show phenotypes associated with actin dynamics regulation. For example, Nullo is a known regulator of actin-myosin stability and has been proposed to affect actin-actin and actin-membrane interactions at the cortex, suggesting a role in cortical remodeling during actomyosin ring contraction [62, 63].

In summary, our understanding of the mechanisms that trigger cell wound repair remain incomplete, but here we show functional translation is essential for initiating a normal and processive wound repair process, suggesting that the first responders are likely mRNA and protein already present in the cell. While transcription is not immediately necessary in the *Drosophila* cell wound model, it is needed for the repair process. The requirement for insulin

signaling in the single cell wound repair context highlights the conservation of repair mechanisms employed. Given its prominence in the single cell, as well as multicellular (tissue), repair pathways, it is not surprising that impaired insulin signaling leads to major wound repair defects in diseases such as diabetes where chronic wounds are symptomatically observed. As many of the top up- and down- regulated genes we identified are evolutionarily conserved genes, but of currently unknown function, the challenge for the future is to determine their roles in normal cellular maintenance and/or development, in addition to their effects in a cell wound repair context, thereby allowing the establishment of a network of cellular processes involved to better aid in treatments of disease involving wound healing impairments, or in disciplines such as regenerative medicine.

## Materials and methods

### Fly stocks and genetics

Flies were cultured and crossed at 25˚C on yeast-cornmeal-molasses-malt extract medium. The flies used in this study are listed in S2A Table. RNAi lines were driven using the GAL4-UAS system using the maternally expressed driver, Pmatalpha-GAL-VP16V37. All genetic fly crosses were performed at least twice. All RNAi experiments were performed at least twice from independent genetic crosses and ≥10 embryos were examined unless otherwise noted.

A calcium reporter, sqh-GCaMP6s (spaghetti squash driven, green fluorescent protein-based GECI), was generated by swapping the GCaMP6s cassette from 20XUAS-GCaMP6s (BDSC #42749) into the constitutive pSqh5′+3′UTR vector [27].

An actin reporter, sGMCA (spaghetti squash driven, moesin-alpha-helical-coiled and actin binding site bound to GFP reporter) [55] or the Cherry fluorescent equivalent, sChMCA [23], was used to follow wound repair dynamics of the cortical cytoskeleton.

*wimp* (*RpL140*$^{wimp}$) reduces maternal gene expression of a specific subset of genes in the early *Drosophila* embryo [79, 80, 108]. Reduced *chickadee* embryos were obtained from transheterozygous females generated by crossing *chickadee*$^{221}$ to *RpL140*$^{wimp}$.

We attempted InR knockdown in three ways: 1) expressing one shRNA (GL00139) using one maternal-GAL4 driver (BDSC #7063), 2) expressing two shRNAs (HMS03166 and GL00139) using one maternal-GAL4 driver (BDSC #70637063), and 3) expressing one shRNA (GL00139) using one maternal-GAL4 (BDSC #7063) in an *InR*$^{05545}$ heterozygous mutant backgrounds. We achieved only 50% knockdown with approach (1), and no eggs were produced by approach (3). We achieved 87% knockdown with approach (2) and this condition was used for the phenotypic analyses included here.

For the MS2-MCP system [50, 51], female virgins maternally expressing MCP-GFP and Histone-RFP were crossed with males expressing 24xMS2 stem loops and lacZ driven by hunchback P2 enhancer and promoter. F1 embryos (MCP-GFP, Histone-RFP/+; 24xMS2-lacZ/+) at NC9-10 stages were used for imaging where the 24xMS2-lacZ mRNA is contributed zygotically.

Localization patterns and mutant analyses were performed at least twice from independent genetic crosses and ≥10 embryos were examined unless otherwise noted. Images representing the average phenotype were selected for figures.

### Controls for RNAi knockdowns

Since the different RNAi lines examined were not all generated in the same genetic background, we used three controls: GAL4 driver+actin reporter (in *w*- background), GAL4 driver +actin reporter+ChFP RNAi (unrelated non-fly RNAi), and GAL4 driver+actin reporter

+*vermillion* RNAi (unrelated fly RNAi). We find that there is no difference among these three controls in any of the measured wound repair parameters: fold expansion, contraction rate, ring width, and ring intensity (see S1 Fig). Throughout the paper, the control shown is the driver+reporter (in *w-* background). These embryos are derived from a cross between mothers harboring the maternal GAL4 driver (Pmatalpha-GAL-VP16V37) and an actin reporter (sGMCA) and fathers mutant for the *white* ($w^{1118}/w^{1118}$) gene.

## Quantification of mRNA levels in RNAi mutants

To harvest total RNA, 100–150 embryos were collected after a 30 min incubation at 25˚C, treated with TRIzol (Invitrogen/Thermo Fisher Scientific) and then with DNase I (Sigma). 1 µg of total RNA and oligo (dT) primers were reverse transcribed using the iScript gDNA Clear cDNA Synthesis Kit (Bio-Rad). RT-PCR was performed using the iTaq Universal SYBR Green Supermix (Bio-Rad) and primers obtained from the Fly Primer Bank listed on S2B Table. We were unable to identify primer sets that would work for qPCR for Geko, Ama, l(3) neo38, danr, and CG4960.

Each gene in question was derived from two individual parent sets and run in two technical replicates on the CFX96 Real Time PCR Detection System (Bio-Rad) for a total of four samples per gene. RpL32 (RP-49) or GAPDH were used as reference genes and the knockdown efficency (%) was obtained using the ΔΔCq calculation method compared to the control (GAL4 only).

## Embryo handling and preparation

NC4-6 embryos were collected for 30 min at 25˚C, then harvested at room temperature (22˚C). Collected embryos were dechorionated by hand, desiccated for 5 min, mounted onto No. 1.5 coverslips coated with glue, and covered with Series 700 halocarbon oil (Halocarbon Products Corp.) as previously described [23].

## Drug injections

Pharmacological inhibitors were injected into NC4-6 staged *Drosophila* embryos, incubated at room temperature (22˚C) for 5 min, and then subjected to laser wounding. The following inhibitors were used: BAPTA (50 mM; Invitrogen); α-amanitin (1 mg/ml; Sigma-Aldrich); puromycin (10 mg/ml; Sigma-Aldrich); and cycloheximide (1 mg/ml; Sigma-Aldrich). The inhibitors were prepared in injection buffer (5 mM KCl, 0.1 mM NaP pH6.8). Injection buffer alone was used as the control. BAPTA was injected into either the perivitelline space (extracellular) or the embryo (intracellular). We were unable to assess calcium dynamics following BAPTA injection simultaneously into the extra- and intra-cellular spaces, as this led to the immediate death of the injected embryos.

## Laser wounding

All wounds were generated with a pulsed nitrogen N2 micropoint laser (Andor Technology Ltd.) set to 435nm and focused at the lateral surface of the embryo. A circular targeted region of 16x15.5 µm was selected along the lateral midsection of the embryo, and ablation was controlled by MetaMorph software (Molecular Devices). Average ablation time was less than 3 seconds and time-lapse image acquisition was initiated immediately after ablation. Upon ablation, a grid-like pattern is sometimes observed (fluorescent dots within the wound area), as a result of the laser scoring the vitelline membrane that envelops the embryo. This vitelline membrane scoring has no effect on wound repair dynamics.

## Immunostaining of wounded embryos

Embryos (1–2 min post-wounding) were fixed in formaldehyde saturated heptane for 40 min. The vitelline membrane was removed by hand and the embryos were then washed 3 times with PAT [1x PBS, 0.1% Tween-20, 1% bovine serum albumin (BSA), 0.05% azide], then blocked in PAT for 2h at 4˚C. Embryos were incubated with mouse anti-chickadee antibody (chi 1J; 1:10; Developmental Studies Hybridoma Bank) and guinea pig anti-Girdin antibody (1:500; provided by Patrick Laprise) [104] and for 24h at 4˚C. Embryos were then washed 3 times with XNS (1x PBS, 0.1% Tween-20, 0.1% BSA, 4% normal goat serum) for 40 min each. Embryos were incubated with Alexa Fluor 568- and Alexa Fluor 633- conjugated secondary antibodies (1:1000; Invitrogen) overnight at 4˚C. Embryos were washed with PTW (1x PBS, 0.1% Tween- 20), incubated with Alexa Fluor 488-conjugated Phalloidin at 0.005 units/µl (Molecular Probes/Invitrogen, Rockford, IL) at room temperature for 1 h, washed with PTW, and then imaged.

## Live image acquisition

All imaging was done using a Revolution WD systems (Andor Technology Ltd.) mounted on a Leica DMi8 (Leica Microsystems Inc.) with a 63x/1.4 NA objective lens under the control of MetaMorph software (Molecular devices). Images were acquired using a 488 nm, 561 nm, and 633 nm Lasers and Andor iXon Ultra 897 EMCCD camera (Andor Technology Ltd.). All time-lapse images were acquired with 17–20 µm stacks/0.25 µm steps. For GCaMP6s imaging, images were acquired every 15 sec for 5 min and then every 60 sec for 25 min. For single color, images were acquired every 30 sec for 15 min and then every 60 sec for 25 min. For dual green and red colors, images were acquired every 45 sec for 30–40 min.

## Image processing, analysis, and quanitification

Image processing was performed using FIJI software [109]. Kymographs were generated using the crop feature to select ROIs of 5.3 x 94.9 µm. To generate fluorescent profile plots by R, 10 pixel sections across the wound from a single embryo were generated using Fiji as we described previously [52]. For dynamic lineplots, we generated fluorescent profile plots from each timepoint and then concatenated them. The lines represent the averaged fluorescent intensity and gray area is the 95% confidence interval. Line profiles from the left to right correspond to the top to bottom of the images unless otherwise noted. Wound area was manually measured using Fiji and the values were imported into Prism 8.2.1 (GraphPad Software Inc.) to construct corresponding graphs. Figures were assembled in Canvas Draw 6 for Mac (Canvas GFX, Inc.).

Quantification of the width and average intensity of actin ring, wound expansion, and closure rate was performed as follows: the width of actin ring was calculated with two measurement, the ferret diameters of the outer and inner edge of actin ring at 120 sec post-wounding. Using these measurements, the width of actin ring was calculated with (outer ferret diameter–inner ferret dimeter)/2. The average intensity of actin ring was calculated with two measurement. Instead of measuring ferret diameters, we measured area and integrated intensity in same regions as described in ring width. Using these measurements, the average intensity in the actin ring was calculated with (outer integrated intensity—inner integrated intensity)/ (outer area—inner area). To calculate relative intensity for unwounded (UW) time point, average intensity at UW was measured with 50x50 pixels at the center of embryos and then averaged intensity of actin ring at each timepoint was divided by average intensity of UW. Wound expansion was calculated with max wound area/initial wound size. Contraction rate was calculated with two time points, one is $t_{max}$ that is the time of reaching maximum wound area, the

other is t<half that is the time of reaching 50–35% size of max wound since the slope of wound area curve changes after t<half. Using these time points, average speed was calculated with (wound area at $t_{max}$−wound area at t<half)/$t_{max}$-t<half. To quantify the level of PIP3-GFP in the actin ring, we used the same method for the measurement of averaged actin ring intensity at 135 sec post-wounding image. Generation of all graphs and student's t test were performed with Prism 8.2.1 (GraphPad Software Inc.).

## Microarray preparation and processing

Expression profiles were obtained using the FHCRC Fly 12k spotted array (GEO platform, GPL 1908). Embryos, prepared for wounding, were either wounded 8 times or left unwounded, then collected for total RNA extraction. Sample labeling and hybridization protocols were performed as described by Fazzio et al [110]. Specifically, cDNA targets were generated from total RNA using a standard amino-allyl labelling protocol where 30 ug of total RNA from each wounding condition (wounded vs non-wounded) were coupled to either Cy3 or Cy5 fluorophores. Targets were co-hybridized to microarrays for 16 hours at 63C and sequentially washed at room temperature (22C) in: 1 x SSC and 0.03% SDS for 2 mins, 1 x SSC for 2 mins, 0.2 x SSC with agitation for 20 mins, and 0.05 x SSC with agitation for 10 mins. Arrays were immediately centrifuged until dry and scanned using a GenePix 4000B scanner (Molecular Devices, Sunnyvale, CA). Image analysis was performed using GenePix Pro 6.0.

## Microarray analysis

Wounded and non-wounded samples were independently replicated 4 times each at the 0 min and 30 min time point. For each array, spot intensity signals were filtered and removed if the values did not exceed 3 standard deviations above the background signal, if the background subtracted signal was <100 in both channels, or if a spot was flagged as questionable by the GenePix Pro Software. Spot-levels ratios were $\log_2$ transformed and loess normalized using the Bioconductor package *limma* [111]. Differential gene expression between wounded and non-wounded states was determined using the Bioconductor package *limma*, and a false discovery rate (FDR) method was used to correct for multiple testing [112]. Significant differential gene expression was defined as $|\log_2 (ratio)| \geq 0.585$ (± 1.5-fold) with FDR set to 5%. Gene ontology enrichment scores were determined using DAVID with significance based on EASE scores corrected for multiple testing [113, 114]. The microarray datasets are available at GEO (NCBI Gene Expression Omnibus) under accession numbers: GSE39481, GSE39482, and GSE39483.

## TAD analysis

Genes were mapped to previously described TADs [45]. A TAD by up/down regulated gene versus unaffected gene expressed on the microarray contigency table was assembled. Fisher's exact test of independence was used to test the null hypothesis that porportion of differentially expressed genes was different per TAD.

## Gene size analysis

Gene size was determined as the size of the largest expressed transcript per gene (dm5.43 build) expressed on the arrays. The median plus 95% CI was determined using the bootstrap procedure and 1000 iterations.

## Statistical analysis

All statistical analysis was done using Prism 8.2.1 (GraphPad, San Diego, CA). Gene knock-downs were compared to the appropriate control, and statistical significance was calculated using a Student's t-test with $p < 0.05$ considered significant. For comparing control lines, a one way ANOVA was performed and all three pair-wise combinations were compared.

## Supporting information

**S1 Table. Changes in gene expression at 30 minutes post-wounding.** List of all 7977 genes present on the microarray with their fold changes in expression at 30 minutes post-wounding. Upregulated genes are highlighted in green; down-regulated genes are highlighted im red. (XLSX)

**S2 Table. Flies and Primers used in this study.** (**A**) Flies used in this study. (**B**) Primers used for q-PCR in this study.
(XLSX)

**S3 Table. Source Data.** Numerical data for graphs in Figs 2–9, S1 Fig, S2 Fig, and S3 Fig.
(XLSX)

**S1 Fig. Controls for RNAi knockdowns.** (**A-B**') Confocal XY projections of actin dynamics at 1, 3, 10, and 15 mpw from Drosophila NC4-6 embryos co-expressing sGMCA with UAS-RNAi for CherryFP (A-A') and Vermilion (B-B') during cell wound repair. (**C**) Quantification of wound area over time for driver only (sGMCA, 7063/+), mCherry RNAi (sGMCA, 7063/Cherry RNAi), and vermilion RNAi (sGMCA, 7063/Vermilion RNAi). (**D-G**) Quantification of wound expansion (D), contraction rate (E), actin ring intensity (F), and actin ring width (G) in driver only, mCherry RNAi, and Vermilion RNAi. $n \geq 10$. One way ANOVA was performed and all three pair-wise combinations were compared (driver vs mCherry RNAi, driver vs vermilion RNAi, and mCherry RNAi vs vermilion RNAi). ns = not significant. See S3 Table for numerical data.
(TIF)

**S2 Fig. Knockdown of up-regulated genes results in wound over-expansion and abnormal actin dynamics.** (**A-H**) Confocal XY projections of actin dynamics at 1, 3, 10, and 15 mpw from *Drosophila* NC4-6 embryos coexpressing sGMCA and a UAS-RNAi transgene during cell wound repair for Geko[RNAi]/+; sGMCA, 7063/+ (A), CG43693[RNAi]/+; sGMCA, 7063/+ (B), Dpn[RNAi]/+; sGMCA, 7063/+ (C), Link[RNAi]/sGMCA, 7063 (D), Ama[RNAi]/sGMCA, 7063 (E), l(3)neo38[RNAi]/sGMCA, 7063 (F), Danr[RNAi]/sGMCA, 7063 (G) and Dtg[RNAi]/sGMCA, 7063 (H). (**A'-H'**) XY kymographs across the wound areas depicted in (A-H), respectively. Note wound overexpansion (yellow arrows); internal actin accumulation (yellow arrowhead), and remodeling defect/open wound (red arrowhead). (**I-P**) Quantification of wound area over time for (A-H'), respectively. (**Q**) Quantification of RNAi efficiencies for each RNAi mutant background (2 biological and 2 technical replicates were performed). Error bars represent ± SEM; $n \geq 10$. See S3 Table for numerical data. Scale bars: 20 μm.
(TIF)

**S3 Fig. Knockdown of down-regulated genes results in wound over-expansion and abnormal actin dynamics.** (**A-I**) Confocal XY projections of actin dynamics at 1, 3, 10, and 15 mpw from *Drosophila* NC4-6 embryos coexpressing sGMCA and a UAS-RNAi transgene during cell wound repair for CG3652[RNAi]/+; sGMCA, 7063/+ (A), NacAlpha [RNAi]/+; sGMCA, 7063/+ (B), Pex11[RNAi]/+; sGMCA, 7063/+ (C), Cyp6a19[RNAi]/sGMCA, 7063 (D), Cyp6a9[RNAi]/sGMCA, 7063 (E), ReepB[RNAi]/sGMCA, 7063 (F), P32[RNAi]/sGMCA, 7063 (G), RpL23[RNAi]/

sGMCA, 7063 (H), and CG145453[RNAi]/sGMCA, 7063 (I). (**A'-I'**) XY kymographs across the wound areas depicted in (A-I), respectively. Note wound overexpansion (yellow arrows); internal actin accumulation (yellow arrowhead), and remodeling defect/open wound (red arrowhead). (**J-R**) Quantification of wound area over time for (A-I'), respectively. Error bars represent ± SEM; n ≥ 10. See S3 Table for numerical data. Scale bars: 20 μm.
(TIF)

**S1 Video. Both intracellular and extracellular calcium sources contribute to the robust calcium response observed upon wounding in the *Drosophila* cell wound model.** (A-D) Time-lapse confocal xy images from Drosophila NC4-6 staged embryos expressing GCaMP6s: control (no injection) (A), buffer injection (D), extracellular BAPTA injection (C), and intracellular BAPTA injection (D). Dynamic smoothened fluorescence intensity profiles (arbitrary units) derived from averaged fluorescence intensity values over a 10 pixel width across the wound area in each timepoint are shown below the image. Gray area represents the 95% CI. Time post-wounding is indicated. UW: unwounded.
(AVI)

**S2 Video. Translation and transcription are needed for different aspects of cell wound repair.** (A-D) Time-lapse confocal xy images from Drosophila NC4-6 staged embryos expressing an actin marker (sGMCA): control (buffer only) (A), alpha-amanitin injected (D), puromycin injected (C), and cycloheximide injected (D). Dynamic smoothened fluorescence intensity profiles (arbitrary units) derived from averaged fluorescence intensity values over a 10 pixel width across the wound area in each timepoint are shown below the image. Gray area represents the 95% CI. Time post-wounding is indicated. UW: unwounded.
(AVI)

**S3 Video. Knockdown of up-regulated genes results in wound over-expansion and abnormal actin dynamics.** (A-H) Time-lapse confocal xy images from Drosophila NC4-6 staged embryos expressing an actin marker (sGMCA): control (w[1118]/+; sGMCA, 7063/+) (A), ImpL2[RNAi]/+; sGMCA, 7063/+ (B), EGFR[RNAi]/+; sGMCA, 7063/+ (C), Gp150[RNAi]/sGMCA, 7063 (D), Inx3[RNAi]/sGMCA, 7063 (E), Thor[RNAi]/sGMCA, 7063 (F), Jbug[RNAi]/sGMCA, 7063 (G), Nullo[RNAi]/sGMCA, 7063 (H). Dynamic smoothened fluorescence intensity profiles (arbitrary units) derived from averaged fluorescence intensity values over a 10 pixel width across the wound area in each timepoint are shown below the image. Gray area represents the 95% CI. Time post-wounding is indicated. UW: unwounded.
(AVI)

**S4 Video. Knockdown of down-regulated genes results in wound over-expansion and abnormal actin dynamics.** (A-G) Time-lapse confocal xy images from Drosophila NC4-6 staged embryos expressing an actin marker (sGMCA): CG31075[RNAi]/+; sGMCA, 7063/+ (A), GstD1[RNAi]/+; sGMCA, 7063/+ (B), dhd[RNAi]/+; sGMCA, 7063/+ (C), Exu[RNAi]/+; sGMCA, 7063/+ (D), CG4960[RNAi]/+; sGMCA, 7063/+ (E), GstD2[RNAi]/+; sGMCA, 7063/+ (F), CG1598[RNAi]/+; sGMCA, 7063/+ (G). Dynamic smoothened fluorescence intensity profiles (arbitrary units) derived from averaged fluorescence intensity values over a 10 pixel width across the wound area in each timepoint are shown below the image. Gray area represents the 95% CI. Time post-wounding is indicated. UW: unwounded.
(AVI)

**S5 Video. Actin dynamics of IIS pathway mutants.** (A-I) Time-lapse confocal xy images from Drosophila NC4-6 staged embryos expressing an actin marker (sGMCA): *insulin-like peptide 4[1]* (*Ilp4[1]*; A), InR[RNAi(1)]/+; InR[RNAi(2)]/sGMCA, 7063 (B), Chico[RNAi]/sGMCA, 7063

(C), Pi3K21B[RNAi]/sGMCA, 7063 (D), Akt1[RNAi]/sGMCA, 7063 (E), FoxO[RNAi]/sGMCA, 7063 (F), Reptor[RNAi]/sGMCA, 7063 (G), Girdin[RNAi]/+; sGMCA, 7063/+ (H), and sGMCA; *chicka-dee[221]*/+ sGMCA, *wimp*/+ (reduced chickadee) (I). Dynamic smoothened fluorescence intensity profiles (arbitrary units) derived from averaged fluorescence intensity values over a 10 pixel width across the wound area in each timepoint are shown below the image. Gray area represents the 95% CI. Time post-wounding is indicated. UW: unwounded.
(AVI)

## Acknowledgments

We thank Ryan Basom, Patrick Laprise, Scott Somers, the Bloomington Stock Center, the Kyoto Stock Center, the Harvard Transgenic RNAi Project, the Vienna Drosophila RNAi Center, the Drosophila Genomics Resource Center, and the Developmental Studies Hybridoma Bank for advice, antibodies, DNAs, flies, and other reagents used in this study.

## Author Contributions

**Conceptualization:** Mitsutoshi Nakamura, Jeffrey M. Verboon, Maria Teresa Abreu-Blanco, Raymond Liu, Susan M. Parkhurst.

**Data curation:** Jeffrey M. Verboon, Jeffrey J. Delrow, Susan M. Parkhurst.

**Formal analysis:** Mitsutoshi Nakamura, Jeffrey M. Verboon, Jeffrey J. Delrow.

**Funding acquisition:** Susan M. Parkhurst.

**Investigation:** Mitsutoshi Nakamura, Jeffrey M. Verboon, Tessa E. Allen, Maria Teresa Abreu-Blanco, Raymond Liu, Andrew N. M. Dominguez, Susan M. Parkhurst.

**Supervision:** Susan M. Parkhurst.

**Validation:** Mitsutoshi Nakamura, Jeffrey M. Verboon, Tessa E. Allen, Maria Teresa Abreu-Blanco, Raymond Liu, Jeffrey J. Delrow, Susan M. Parkhurst.

**Visualization:** Mitsutoshi Nakamura, Jeffrey M. Verboon, Tessa E. Allen, Andrew N. M. Dominguez, Susan M. Parkhurst.

**Writing – original draft:** Mitsutoshi Nakamura, Jeffrey M. Verboon, Susan M. Parkhurst.

**Writing – review & editing:** Tessa E. Allen, Maria Teresa Abreu-Blanco, Raymond Liu, Andrew N. M. Dominguez, Jeffrey J. Delrow.

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
