## [Decision Letter · Decision Letter 0]

31 Aug 2020

Dear Dr Parkhurst,

Thank you very much for submitting your Research Article entitled 'Autocrine insulin pathway signaling regulates actin dynamics in cell wound repair' to PLOS Genetics. Your manuscript was fully evaluated at the editorial level and by independent peer reviewers. The reviewers appreciated the attention to an important topic but identified some aspects of the manuscript that should be improved.

We therefore ask you to modify the manuscript according to the review recommendations before we can consider your manuscript for acceptance. Your revisions should address the specific points made by each reviewer.

[LINK]

Yours sincerely,

Denise J. Montell

Associate Editor

PLOS Genetics

Gregory P. Copenhaver

Editor-in-Chief

PLOS Genetics

Dear Susan,

Your manuscript has been evaluated by three reviewers, whose verbatim comments are included. As you can see, there is considerable enthusiasm for this study. Reviewers 1 and 2 request only minor revisions whereas Reviewer 3 has more substantial questions about some technical points as well as the overall advance this represents for the field. In particular the question about the same image appearing twice in one figure is of some concern. We believe the manuscript could be improved by revising it in response to these critiques. Please focus on those questions and comments that concern the validity and interpretation of the results presented, rather than extensive experiments that substantially expand the scope of the study. We look forward to receiving a revised manuscript that addresses all of the reviewers' concerns.

Reviewer's Responses to Questions

**Comments to the Authors:**

Reviewer #1: Review of "Autocrine insulin pathway signaling regulates actin dynamics in cell wound repair" by Nakamura et al.

The repair of damaged cells is a basic biological process that is conserved across phyla ranging from yeast to humans. Further, it is of profound clinical importance in that its failure is associated with and/or causative of several common human diseases including muscular dystrophies and diabetes. And yet, this cell repair is very poorly understood, particularly when compared to other fundamental cell processes such as the cell division cycle or secretion. There are several reasons why our grasp of cell repair is so rudimentary, but arguably one of the most fundamental is that this process has yet to be subject to investigation using all of the tools now available in genetically tractable model organisms.

From this standpoint alone, the study by Nakamura et al. is well worth the price of admission. The authors have exploited transcriptomic analysis of wounded Drosophila syncytial embryos to identify a wealth of candidate players in cell repair, and directly confirmed the importance of many of these using RNAi and genetic manipulations. Thus, those who study cell repair have new specific targets to investigate.

But the importance of this study goes well beyond the identification of new cell repair participants. One of the most surprising findings is actually very general: repair in this system requires protein synthesis. Most in this field assume that because many features of cell repair are extremely rapid (for example, resealing of the damaged plasma membrane is generally thought to be complete within a few seconds while the cortical cytoskeletal response commences within ~30-60s) repair mechanisms must be mostly, if not exclusively post-translational. However, Nakamura et al. show that this is clearly not the case. Profound deficits in the cortical cytoskeletal response to damage ensue when embryos are wounded in the presence of protein synthesis inhibitors. Further studies will be required to determine which proteins are subject to immediate translational upregulation following damage, but simply knowing that such exist sets the stage for such experiments.

The second general finding, that transcription is needed for the later stages of healing, is perhaps less surprising, but also important, in that the field has been obsessed with the early events. However, this obsession is misplaced in that the later events of repair are just as likely to be clinically important as the early ones. By rigorously demonstrating the importance of transcription for later repair events, and then identifying several players specifically required for the later events, the authors call attention to both.

The authors also show that participants in insulin signaling are upregulated following cell damage and, even more remarkably, that the insulin signaling pathway is actually triggered by damage. Previous work has demonstrated that cells from diabetic patients are deficient in repair and that induction of the diabetic state by simply incubating isolated cells in excess glucose for several days is enough to induce a healing deficit (). However, it was assumed that the loss of healing capacity reflected systemic changes in the manipulated cells rather than a direct participation of insulin signaling in the healing process.

I could go on, but the point should be clear by now: this study has all of the earmarks of a classic.

Only a few concerns, quite minor, deserve mention:

1. In Figure 2, the demonstrations that the inhibitors are having the expected effects on transcription and translation (currently Figs. 2H-K) should probably come before the analysis of their effects on healing (currently Figs. 2A-D).

2. As noted above, deficits in single cell repair have been demonstrated for cells from diabetic mice or cells induced to become diabetic (Diabetes 2011. 60:3034-3042). This citation should be included since it strengthens the connection of the current study to other cell repair models.

3. Localization of PIP3 to vertebrate cell wounds has previously been described (Mol. Biol. Cell. 2014. 25:1867-1876). This citation should be included since it strengthens the connection of the current study to other cell repair models.

4. This is up to the authors, but they may wish to call attention to the fact that there is precedent for fast acting changes in translation being used to control other Rho GTPase-dependent processes, such as axon guidance (Nature 2005. 436:1020-1024).

Reviewer #2: This is an interesting paper by Parkhurst lab and study the effect of laser wounding in syncytial blastoderm embryos in Drosophila. They report that early response to wounding does not involve a transcriptional response, but rather translational one. Given that the development in syncytial embryos is mostly driven by maternal transcripts, this is maybe not surprising result. But, zygotic transcription is initiated in late steps of wound repair suggesting that dormant transcription is activated by wounding. To learn more about the transcriptional response, a microarray analysis was performed and 253 up- and down-regulated genes were identified. 15 up and 16 down-regulated genes were tested by RNAi and all disrupted mportant steps of the repair process. One exciting outcome of this analysis is that the insulin pathway is an essential component of repair. This is achieved through IIS control of the actin regulators Chickadee and Girdin.

Overall, I enjoyed reviewing this paper. It contains a lot of work and makes 3 important points: that cellular wound repair is not dependent on transcriptional activity to initiat repair, that dormant transcription is activated in response to wounds and that the insulin signaling pathway is an essential component of the repair process. The paper is well written and the figures look nice. Fig. 9 nicely illustrates and summarizes the model. I only have a few suggestions for improvement.

Do the RNAi lines tested in fig. 3-5 affect embryogenesis in any way?

What is the control in these figures? I prefer an unrelated RNAi line such as luciferase, GFP or RFP RNAi.

Outline in mor detail for the non-experts why the size of the genes is important.

Reviewer #3: Comments for the authors:

The manuscript Nakamura et al attempts to give a molecular insight into wound healing process employing the fly embryo. The authors carry out microarray analysis between normal and wounded embryos to identify the genes that contribute to the wound healing process. As the authors didn’t observe any significant differences in the gene expression between the control and wounded embryos just after wounding, they propose that initial recovery is aided by events that are mostly translational. The authors have nicely supported the above conclusion using the combination of MCP-MS2 system and drug treatments. Subsequently the authors identified several differentially expressed genes 30 minutes post wounding and carried out preliminary validation of some of identified targets in wound healing assay. Subsequently the authors narrowed down on Insulin signalling to get a molecular insight into how it affects wound healing process. Given that there are quite a few reports that give some insight into how Insulin signalling affects wound healing, I am not sure if the data presented in this regard is a significant advance in the field of wound healing.

Concerns1:

To validate their microarray data, the authors have tested a range of parameters of healing wounds like wound area, contraction rate, actin dynamics etc in different genetic backgrounds for the duration (400s- 500s post wounding) when the post transcriptional events would be more pronounced than the transcriptional input. Thus it is not clear how much of transcriptional component of the shortlisted gene regulates healing process during this duration. In other words, down regulation of ImpL2 exhibits an altered actin dynamics right at 3 minutes post wounding, which I would believe must be because of the lower abundance of the protein at that time point and may not be directly connected with observed transcriptional up regulation 30 minutes post wounding. Secondly, can the authors delineate when the transcriptional components start playing an active role in healing process after wounding.

Concern 2:

Given that Foxo and Raptor are the negative regulator of Insulin signalling, it is very difficult to comprehend the behavior of fold expansion, contraction rate, actin ring intensity of Foxo & Raptor depleted embryos with that of positive regulators of Insulin signalling like InR, dilp4, chico depleted embryos in wound healing. Since depleting both the positive and negative regulators of Insulin signalling impedes or slows the wound healing process, an explanation supported by experiments would be helpful.

Concern 3:

The genes that were found to be down regulated upon wounding in the microarray data, were further down regulated using RNAi for functional analysis. I am not sure how these experiments would give meaningful insight into wound healing, however authors refer to some possibilities in the discussion section. It would be better if these hypotheses/ speculation could be supported by experiments or microarray data analysis by looking at the genes that are already implicated in wound healing so far.

.

Concern 4:

In Fig 6B-B'' and 6C-C'', the time elapsed post wounding is not mentioned. Going by the actin ring formation in the control samples (Fig 6B), they seems comparable to the 3 minute post wounding phenotype observed in Fig 2A and Fig 3A. If indeed they are in that timepoint, then it can be said with confidence that the transcriptional up regulation of ImpL2 clearly does not regulate the recruitment of PIP3 in the actomyosin ring region. And that would give rise to two obvious questions:

a) How is Insulin signalling aiding PIP3 recruitment on the actomyosin ring?

b) What purpose is the observed transcriptional up regulation of ImpL2 actually serving?

Concern 5:

Some experiments that give molecular sight into how Insulin signalling regulates actin dynamics during wound healing will be helpful for supporting the model proposed by the authors.

Concern 6:

The authors propose that the Calcium surge probably causes the autocrine activation of the Insulin signalling pathway and thereby mediating the wound healing. The cited references [17, 18, 73, 74] are in either sea urchin or mammalian cell lines. The model will hold valid if the authors can also experimentally support that laser ablation causes Calcium surge, say by employing a genetically encoded calcium reporter. This might be a very important experiment to support the model. Also of interest is the source of calcium. Is calcium already present in the peri-vitelline space or is it released from the laser ablated portion of the tissue, which can be discussed? Therefore it might be important to show some degree of experimental data to hint that Calcium mediated entry might be causing the release of ILPs and thereby resulting in an autocrine IlS pathway activation. Without any substantial evidence, the inclusion of calcium in the model seems a very weak link to support the activation of IlS signalling pathway. Secondly how do the authors rule out the possibility that the laser ablation itself causes damage to the pre-packed ILPs in vesicles, thereby releasing ILPs and subsequent InR activation.

Concern 7:

The images in the panel Figure 3 F and 3H, seem to be the "exactly" same image other than the differences in contrast/brightness. I would request the authors to check it.

Concern 8:

Have the authors attempted rescuing the phenotype of wounding by over-expressing the actin regulators, Girdin or Chickadee, in the background where Insulin signalling is down regulated ?

Concern 9:

The discussion section may be modified to compare and contrast the present results with that of the previous ones so far including that the one reported by Kakanj et al 2016.

Concern 10:

I couldn't get an idea as what is exactly meant by fold expansion in the assays.

**Have all data underlying the figures and results presented in the manuscript been provided?**

Reviewer #1: Yes

Reviewer #2: Yes

Reviewer #3: Yes

PLOS authors have the option to publish the peer review history of their article (what does this mean?). If published, this will include your full peer review and any attached files.

Reviewer #1: **Yes: **William Bement

Reviewer #2: No

Reviewer #3: No

---

## [Editor Report · Decision Letter 1]

9 Oct 2020

Dear Dr Parkhurst,

We are pleased to inform you that your manuscript entitled "Autocrine insulin pathway signaling regulates actin dynamics in cell wound repair" has been editorially accepted for publication in PLOS Genetics. Congratulations!

Yours sincerely,

Denise J. Montell

Associate Editor

PLOS Genetics

Gregory P. Copenhaver

Editor-in-Chief

PLOS Genetics

Comments from the reviewers (if applicable):

As you have responded throughly to the reviewers' comments, we are happy to accept the manuscript for publication. Thanks for submitting this interesting study!

**Data Deposition**

http://datadryad.org/submit?journalID=pgenetics&manu=PGENETICS-D-20-01146R1

**Press Queries**

---

## [Editor Report · Acceptance letter]

28 Nov 2020

PGENETICS-D-20-01146R1 

Autocrine insulin pathway signaling regulates actin dynamics in cell wound repair 

Dear Dr Parkhurst, 

We are pleased to inform you that your manuscript entitled "Autocrine insulin pathway signaling regulates actin dynamics in cell wound repair" has been formally accepted for publication in PLOS Genetics! Your manuscript is now with our production department and you will be notified of the publication date in due course.

With kind regards,

Nicola Davies

PLOS Genetics

On behalf of:
